# Development and structural basis of a two-MAb cocktail for treating SARS-CoV-2 infections

Chao Zhang[1,7], Yifan Wang[2,7], Yuanfei Zhu[3,7], Caixuan Liu[2,7], Chenjian Gu[3,7], Shiqi Xu[1,7], Yalei Wang[1], Yu Zhou[1], Yanxing Wang[2,4], Wenyu Han[2], Xiaoyu Hong[2], Yong Yang[1], Xueyang Zhang[1], Tingfeng Wang[1], Cong Xu[2], Qin Hong[2], Shutian Wang[2], Qiaoyu Zhao[2], Weihua Qiao[1], Jinkai Zang[1], Liangliang Kong[5], Fangfang Wang[5], Haikun Wang ✪ [1], Di Qu[3,6], Dimitri Lavillette ✪ [1], Hong Tang[1], Qiang Deng ✪ [3✉], Youhua Xie ✪ [3✉], Yao Cong ✪ [2,4✉] & Zhong Huang ✪ [1✉]

The ongoing pandemic of coronavirus disease 2019 (COVID-19) is caused by severe acute respiratory syndrome coronavirus 2 (SARS-CoV-2). Neutralizing antibodies against SARS-CoV-2 are an option for drug development for treating COVID-19. Here, we report the identification and characterization of two groups of mouse neutralizing monoclonal antibodies (MAbs) targeting the receptor-binding domain (RBD) on the SARS-CoV-2 spike (S) protein. MAbs 2H2 and 3C1, representing the two antibody groups, respectively, bind distinct epitopes and are compatible in formulating a noncompeting antibody cocktail. A humanized version of the 2H2/3C1 cocktail is found to potently neutralize authentic SARS-CoV-2 infection in vitro with half inhibitory concentration (IC50) of 12 ng/mL and effectively treat SARS-CoV-2-infected mice even when administered at as late as 24 h post-infection. We determine an ensemble of cryo-EM structures of 2H2 or 3C1 Fab in complex with the S trimer up to 3.8 Å resolution, revealing the conformational space of the antigen–antibody complexes and MAb-triggered stepwise allosteric rearrangements of the S trimer, delineating a previously uncharacterized dynamic process of coordinated binding of neutralizing antibodies to the trimeric S protein. Our findings provide important information for the development of MAb-based drugs for preventing and treating SARS-CoV-2 infections.

[1] CAS Key Laboratory of Molecular Virology & Immunology, Institut Pasteur of Shanghai, Chinese Academy of Sciences, University of Chinese Academy of Sciences, Shanghai, China. [2] State Key Laboratory of Molecular Biology, National Center for Protein Science Shanghai, Shanghai Institute of Biochemistry and Cell Biology, Center for Excellence in Molecular Cell Science, Chinese Academy of Sciences, University of Chinese Academy of Sciences, Shanghai, China. [3] Key Laboratory of Medical Molecular Virology (MOE/NHC/CAMS), Department of Medical Microbiology and Parasitology, School of Basic Medical Sciences, Shanghai Medical College, Fudan University, Shanghai, China. [4] Shanghai Science Research Center, Chinese Academy of Sciences, 201210 Shanghai, China. [5] The National Facility for Protein Science in Shanghai (NFPS), 201210 Shanghai, China. [6] BSL-3 Laboratory of Fudan University, School of Basic Medical Sciences, Shanghai Medical College, Fudan University, Shanghai, China. [7] These authors contributed equally: Chao Zhang, Yifan Wang, Yuanfei Zhu, Caixuan Liu, Chenjian Gu, Shiqi Xu. ✉email: qdeng@fudan.edu.cn; yhxie@fudan.edu.cn; cong@sibcb.ac.cn; huangzhong@ips.ac.cn

The ongoing pandemic of coronavirus disease 2019 (COVID-19) is caused by a newly identified coronavirus named severe acute respiratory syndrome coronavirus 2 (SARS-CoV-2; formerly designated 2019-nCoV)[1–3]. Individuals infected with SARS-CoV-2 may develop severe respiratory manifestations and even death with a fatality rate of ~5% (ref. [4]). Extensive efforts have been made to rapidly develop vaccines and therapies against SARS-CoV-2 (refs. [5,6]).

SARS-CoV-2 is an enveloped, single-stranded, positive-sense RNA virus belonging to the *Betacoronavirus* genus within the *Coronaviridae* family[7]. Spike (S) protein protrudes from the surface of the spherical virions and mediates virus entry into host cells. It consists of an ectodomain comprised of the S1 receptor-binding subunit and the S2 membrane fusion subunit, a transmembrane domain, and a short intracellular tail. The S1 subunit mainly consists of the N-terminal domain (NTD) and the C-terminal domain (CTD). The CTD directly engages the cellular receptor, human angiotensin-converting enzyme 2 (ACE2), and functions as the receptor-binding domain (RBD)[1,8–11]. RBD is composed of the core structure and the receptor-binding motif (RBM; residues 439–506) that is responsible for directly engaging the ACE2 receptor[10,11]. The S protein on the SARS-CoV-2 virion forms trimers. It has been recently shown that two different states of the trimeric SARS-CoV-2 S glycoprotein exist, called "closed" (receptor-inaccessible) and "open" (receptor-accessible) states[12–17]. In the closed state, all three RBDs are in down conformation, whereas for the open state only a single RBD is in up position, which is thought to be less stable[12,13,17].

Neutralizing antibodies play a major role in the antiviral immunity and have been shown to be a viable option for developing therapies against viral infections[18,19]. Recently, human monoclonal antibodies (MAbs) with neutralization effects on SARS-CoV-2 have been identified by a number of groups[20–31]. These MAbs possess varied neutralization potency and receptor blocking ability. Among them, some MAbs were found to be therapeutic effective in mouse models of SARS-CoV-2 infection when the antibody treatment was initiated no later than 12 h post infection (h.p.i.)[20,21,26,29]. It remains unknown whether these MAbs, when given at a delayed time point after virus challenge, will still be efficacious. The possibility of antibody resistance due to the emergence of virus escape mutants is another concern for developing MAb-based treatment. For example, a recent study reports that escape mutants rapidly appeared following in vitro passaging of replicating VSV-SARS-CoV-2-S virus in the presence of individual MAbs, whereas treatment with a noncompeting, two-MAb cocktail did not generate escape mutants[30]. Therefore, it is important to discover more powerful anti-SARS-CoV-2 MAbs for formulating an MAb-based therapy with an extended therapeutic window and minimized risk of developing drug resistance.

The epitopes of the newly identified anti-SARS-CoV-2-neutralizing MAbs were found to be located on the S protein, particularly its RBD, as defined by structural studies of MAbs in complex with recombinant RBD protein, using either crystallization or cryo-EM approaches[20,24,27,28], suggesting that blockade of the interaction between RBD and the ACE2 receptor is the main mechanism for MAb-mediated neutralization. However, these studies still limit us from comprehensive understanding of the structural basis for neutralizing MAb binding and function, primarily because the S protein on the SARS-CoV-2 virion forms trimers that may exist in at least two different conformational states ("closed" and "open"), with distinct positioning and conformation of their three RBD subunits[12,13]. Some important questions remain unaddressed, e.g., the occupancy of antibodies and the global effect of antibody binding in the context of S trimers.

In this study, we identified and comprehensively characterized two groups of mouse anti-SARS-CoV-2-neutralizing MAbs. MAbs 2H2 and 3C1, representing the two antibody groups, respectively, targeted distinct epitopes on RBD and were compatible in formulating a noncompeting antibody cocktail. A humanized version of the 2H2/3C1 cocktail was found to synergistically neutralize authentic SARS-CoV-2 infection in vitro and effectively treat SARS-CoV-2-infected mice when given at as late as 24 h.p.i. Moreover, we captured an ensemble of cryo-EM structures of SARS-CoV-2 S trimer in complex with the Fab of 2H2 or 3C1 up to 3.8 Å resolution, revealing the MAb-triggered stepwise allosteric rearrangements of the S trimer to coordinate the binding of neutralizing antibodies targeting the SARS-CoV-2 RBD, also providing structural basis for MAbs 2H2 and 3C1 as noncompeting antibody cocktail. Our findings provide important information for the development of MAb-based drugs for preventing and treating SARS-CoV-2 infections.

## Results

**Isolation and characterization of SARS-CoV-2-neutralizing MAbs.** We attempted to generate SARS-CoV-2-neutralizing MAbs from mice immunized with a recombinant protein containing the SARS-CoV-2 RBD fused with a C-terminal mouse IgG Fc (RBD-mFc) by using the conventional hybridoma technology. Culture supernatants from the resulting hybridoma clones were screened for RBD binding, blockade of ACE2 binding to immobilized RBD, and neutralization of SARS-CoV-2 pseudovirus (Supplementary Fig. 1). The results showed that 31 hybridoma clones strongly bound SARS-CoV-2 RBD, among which 12 were found to cross-react with the recombinant RBD of SARS-CoV. A total of nine hybridoma clones (#1, #4, #5, #11, #16, #25, #28, #29, and #31) exhibited strong competition with ACE2 for binding to SARS-CoV-2 RBD, and six of them (clones #11, #16, #25, #28, #29, and #31) were able to potently neutralize SARS-CoV-2 pseudovirus (containing a luciferase reporter gene) infection. Consistently, five (#16, #25, #28, #29, and #31) out of the nine ACE2-competing hybridoma clones showed strong neutralizing activity against SARS-CoV-2 pseudovirus containing the GFP reporter, while the other four clones (#1, #4, #5, and #11) exhibited weak inhibitory effect (Supplementary Fig. 2). Based on the results from the two neutralization assays, we selected the five strongly neutralizing MAb clones (#16, #25, #28, #29, and #31), designated 3C1, 2H2, 2G3, 3A2, and 8D3, respectively, for subsequent in-depth studies. Isotyping assay showed that MAb 3A2 is IgG2b, while the other four MAbs belong to IgG1 class (Fig. 1b). The coding sequences for these MAbs were determined and analyzed using IgBLAST[32], and the results showed that antibody variable regions of the five clones were derived from different germline genes (Supplementary Fig. 3), indicating that these five MAbs were distinct clones.

Purified MAbs were firstly evaluated for binding to SARS-CoV-2 RBD and SARS-CoV RBD protein by ELISA. All of the five anti-SARS-CoV-2 MAbs, but not the irrelevant isotype control MAb, dose-dependently bound SARS-CoV-2 RBD with half maximal effective concentration (EC50) ranging from 8.4 to 21.6 ng/mL (Fig. 1a). In addition, MAb 3C1 cross-reacted with SARS-CoV RBD with EC50 of 31.4 ng/mL, whereas the other four anti-SARS-CoV-2 MAbs did not regardless of the antibody concentration (Supplementary Fig. 4a). The MAbs were then assessed for binding affinities to different antigens, including recombinant SARS-CoV-2 RBD, SARS-CoV-2 S trimer, and SARS-CoV RBD, by bio-layer interferometry (BLI) assay. The five anti-SARS-CoV-2 MAbs showed high binding affinities to SARS-CoV-2 RBD with equilibrium dissociation constants ($K_D$) <1 pM

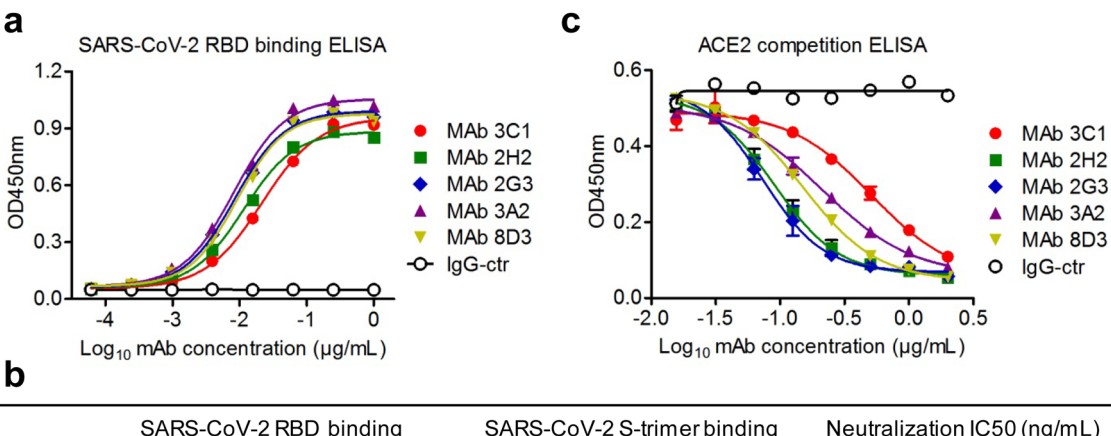

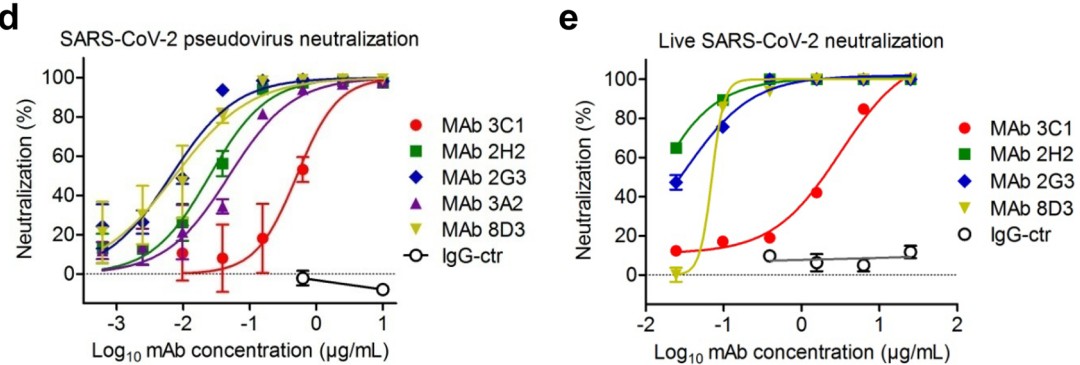

| MAb | Isotype | SARS-CoV-2 RBD binding | | | SARS-CoV-2 S-trimer binding | | | Neutralization IC50 (ng/mL) | |
|---|---|---|---|---|---|---|---|---|---|
| | | KD (nM) | Kon (1/Ms) | Kdis (1/s) | KD (nM) | Kon (1/Ms) | Kdis (1/s) | Pseudovirus | Live virus |
| 3C1 | IgG1 | < 0.001 | $1.2 \times 10^5$ | $< 1.0 \times 10^{-7}$ | < 0.001 | $1.6 \times 10^5$ | $< 1.0 \times 10^{-7}$ | 508 | 3127 |
| 2H2 | IgG1 | < 0.001 | $2.6 \times 10^5$ | $< 1.0 \times 10^{-7}$ | 0.18 | $2.9 \times 10^5$ | $5.1 \times 10^{-5}$ | 25 | 7 |
| 2G3 | IgG1 | < 0.001 | $2.0 \times 10^5$ | $< 1.0 \times 10^{-7}$ | 0.40 | $2.0 \times 10^5$ | $8.0 \times 10^{-5}$ | 7 | 32 |
| 3A2 | IgG2b | < 0.001 | $1.6 \times 10^5$ | $< 1.0 \times 10^{-7}$ | < 0.001 | $1.6 \times 10^5$ | $< 1.0 \times 10^{-7}$ | 49 | Not tested |
| 8D3 | IgG1 | < 0.001 | $2.1 \times 10^5$ | $< 1.0 \times 10^{-7}$ | 0.005 | $4.3 \times 10^5$ | $2.0 \times 10^{-6}$ | 7 | 71 |

**Fig. 1 Binding properties, receptor-binding inhibitory activity, and neutralization activity of the MAbs. a** Reactivities of anti-SARS-CoV-2 MAbs to the SARS-CoV-2 RBD measured by ELISA. Data are mean ± SEM of triplicate wells. Zika virus (ZIKV)-specific MAb 5F8 served as IgG1 isotype control (IgG-ctr) and was used as a control in all subsequent experiments. **b** Isotypes, binding affinities, and neutralization activity of the MAbs. Binding affinities of the MAbs to immobilized SARS-CoV-2 RBD and S trimer were determined by bio-layer interferometry (BLI). **c** Competition between the MAbs and ACE2 for binding to SARS-CoV-2 RBD was measured by ELISA. Biotinylated ACE2-hFc fusion protein was tested for the ability to bind to immobilized RBD in presence of the MAbs, and the signal was detected using HRP-conjugated streptavidin. Data are mean ± SEM of triplicate wells. **d** The MAbs neutralized SARS-CoV-2 pseudovirus infection in vitro. The purified MAbs were fourfold serially diluted and evaluated for neutralization of murine leukemia virus (MLV) pseudotyped with SARS-CoV-2 spike protein. Luciferase activity was measured 2 days after infection. Results shown are representative of two independent experiments. Data are expressed as mean ± SEM of five replicate wells. **e** The MAbs neutralized authentic SARS-CoV-2 infection in vitro. Serially diluted purified MAbs were subjected to live SARS-CoV-2 virus neutralization assay. After 48 h culture, viral RNA in cells were detected by RT-qPCR. Data are mean ± SEM of triplicate wells.

and to SARS-CoV-2 S trimer with $K_D$ values ranging from <1 pM to 0.4 nM (Fig. 1b and Supplementary Fig. 4b, c). In addition, the MAb16-3C1 also displayed high binding affinity toward SARS-CoV RBD with $K_D$ of 1.0 nM (Supplementary Fig. 4d).

Next, we determined the ability of the MAbs to block the interaction between SARS-CoV-2 RBD and ACE2 receptor (Fig. 1c). The five anti-SARS-CoV-2 MAbs, but not the isotype control antibody, were found to dose-dependently inhibit ACE2 binding to RBD, with half inhibitory concentrations (IC50) ranging from 0.074 to 0.510 μg/mL.

To evaluate the neutralization potency of the MAbs, we firstly performed neutralization assays with SARS-CoV-2 pseudovirus. The five anti-SARS-CoV-2 MAbs were found to potently neutralize SARS-CoV-2 pseudovirus infection of human ACE2-

overexpressing HEK 293 T cells (293T-hACE2) with IC50s determined to be 0.508 μg/mL for 3C1, 0.025 μg/mL for 2H2, 0.007 μg/mL for 2G3, 0.049 μg/mL for 3A2, and 0.007 μg/mL for 8D3 (Fig. 1b, d).

MAbs 3C1, 2H2, 2G3, and 8D3, were further assessed for neutralization of authentic SARS-CoV-2 infection of VeroE6 cells. Results from both qRT-PCR and immunostaining assays demonstrated that these anti-SARS-CoV-2 MAbs could efficiently neutralize authentic SARS-CoV-2 infection (Fig. 1e and Supplementary Fig. 5). The IC50 values for 2H2, 2G3, 8D3, and 3C1, were determined to be 0.007, 0.032, 0.071, and 3.127 μg/mL, respectively (Fig. 1b, e). Apparently, among the five neutralizing MAbs, 2H2 is the strongest and 3C1 the weakest in terms of their neutralization potency.

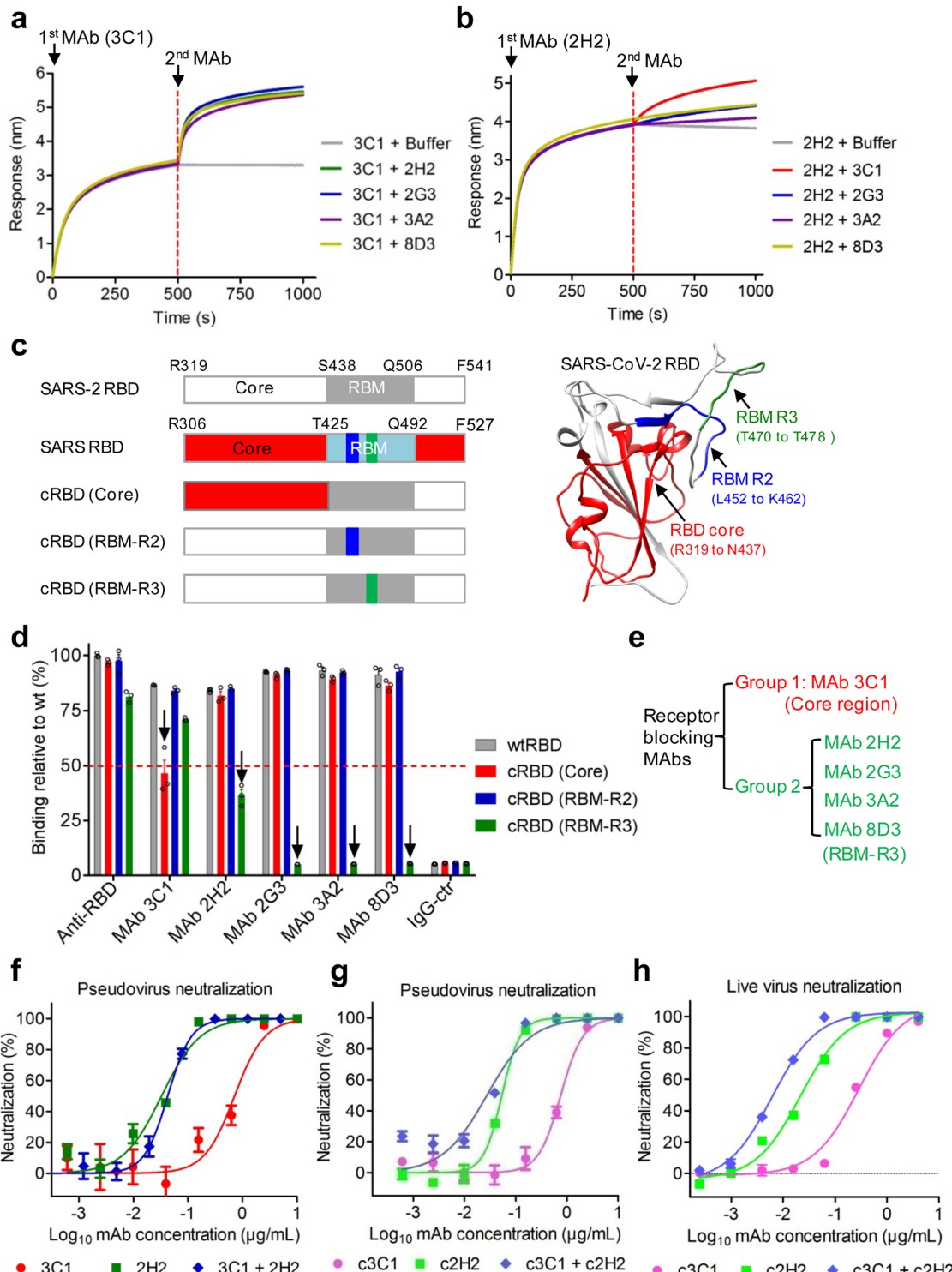

**Selection and humanization of a noncompeting two-antibody cocktail.** Combined use of two or more antiviral MAbs targeting distinct epitopes may increase therapeutic efficacy and reduce the risk of acquiring drug resistance[30]. Therefore, we attempted to identify a noncompeting MAb pair from the five individual neutralizing MAbs. BLI-based antibody competition assays were performed. In the first experiment, immobilized SARS-CoV-2 RBD was saturated with 3C1 (first antibody) and then allowed to interact with a second antibody or dissociate in the buffer. As shown in Fig. 2a, 3C1 (100 nM) hardly dissociated in the buffer, showing high binding affinity; incubation with each of the other

four MAbs led to significant increases in BLI signals, suggesting that 3C1 targets an epitope distinct from the binding sites of the other MAbs. In the second experiment, immobilized SARS-CoV-2 RBD was saturated with MAb 2H2, followed by incubation with a second MAb or dissociation in the buffer. The result showed that 3C1 produced the highest additional BLI signal (Fig. 2b), implicating that 3C1 and 2H2 recognize distinct epitopes on RBD; in contrast, incubation with 2G3, 3A2, or 8D3 resulted in only a slight increase in BLI signals, suggesting that the epitopes of these three antibodies may overlap with that of the MAb 2H2.

**Fig. 2 Antibody competition, epitope mapping, and generation of antibody cocktail. a, b** Antibody binding competition assay. Antibody competition for binding to SARS-CoV-2 RBD was measured by BLI. Immobilized RBD was first saturated with the first antibody MAb 3C1 (**a**) or MAb 2H2 (**b**), and then a second MAb (MAb names were shown after the plus sign) or dissociation buffer (control) was added and allowed to react with the RBD. **c** Diagrams of chimeric RBD mutants (cRBD). cRBD (core), the N-terminal residues R319 to N437 of core region in the SARS-CoV-2 RBD were mutated into the corresponding part of SARS-CoV. cRBD (RBM-R2) and cRBD (RBM-R3), residues L452 to K462, and residues T470 to T478 of RBM region in the SARS-CoV-2 RBD were separately substituted by the corresponding residues of SARS-CoV. The positions of the mutated amino acids are shown in the wild-type RBD crystal structure (PDB: 6M0J; right panel). **d** Reactivities of the MAbs to wild-type (wt) and mutant SARS-CoV-2 RBD proteins measured by ELISA. RBD-mFc immune sera (anti-RBD) served as positive control. The downward arrow indicates that substitutions in RBD mutants significantly reduced the binding of the MAbs compared to wild-type RBD. The reactivity level of wild-type SARS-CoV-2 RBD and anti-RBD sera was set to 100%, and the red dashed line represents 50% reduction relative to wild type. Data are mean ± SEM of triplicate wells. Each symbol represents one well. **e** Grouping of the MAbs. Group 1, MAb16-3C1; group 2, the other MAbs. Antibody epitopes were shown in brackets. **f** Neutralization activity of the murine 2H2/3C1 cocktail. 2H2 alone, 3C1 alone, and the 2H2/3C1 (1:1) cocktail were serially diluted and evaluated for neutralization of SARS-CoV-2 pseudovirus. **g** Neutralization activity of the chimeric MAb cocktail against SARS-CoV-2 pseudovirus. c2H2 alone, c3C1 alone, and the c2H2/c3C1 (1:1) cocktail were serially diluted and assessed for neutralization of SARS-CoV-2 pseudovirus. For **f** and **g** luciferase activity was measured 2 days after infection. Data are expressed as mean ± SEM of five replicate wells. **h** Neutralization activity of the chimeric MAb cocktail against authentic SARS-CoV-2. Serially diluted purified MAbs were subjected to live SARS-CoV-2 virus neutralization assay. After 48 h culture, viral RNA in cells were detected by RT-qPCR. Data are mean ± SEM of triplicate wells. For **f–h**, for MAb cocktails the concentration on the x-axis is that of the 2H2 or c2H2 antibody.

To roughly map the antibody epitopes, we designed six chimeric RBD mutants by replacing a domain/fragment of SARS-CoV-2 RBD with the corresponding part of closely related SARS-CoV (details of the variations for each mutant were presented in Supplementary Fig. 6). Three of these RBD mutants were successfully expressed and purified from transfected HEK 293 F cells, whereas the yields of the other mutants were too poor to proceed further. The three expressed chimeric RBD mutants, designated cRBD (core), cRBD (RBM-R2), and cRBD (RBM-R3; Fig. 2c), were compared in parallel with the wild-type SARS-CoV-2 RBD (wtRBD) for antibody binding in ELISA. As shown in Fig. 2d, when MAb 3C1 was used as the detection antibody, the ELISA signal produced by the cRBD (core) mutant was significantly decreased as compared to those of the wtRBD and the other two mutants, suggesting that 3C1 was directed against the core region of RBD. When detected with each of the other four MAbs, the cRBD (Core) and cRBD (RBM-R2) mutants yielded strong reactivity as did the wtRBD; in contrast, the binding activity to the cRBD (RBM-R3) was drastically reduced (for 2H2) or almost completely abolished (for 2G3, 3A2, and 8D3), indicating that the RBM-R3 region (residues T470 to T478) was involved in the recognition of RBD by the four MAbs.

Based on the above binding competition and epitope mapping data, we divided the five MAbs into two antibody groups targeting distinct antigenic sites: group 1 consists of only MAb 3C1 which likely binds the core region of RBD; group 2 is comprised of the other four MAbs whose epitopes involve residues T470 to T478 of the RBM within RBD (Fig. 2e). We selected 3C1 from group 1 and 2H2 from group 2 for pairing. An antibody cocktail was formulated by mixing 3C1 and 2H2 at a ratio of 1:1 and tested for neutralization against SARS-CoV-2 pseudovirus. The neutralization curves for the 2H2/3C1 cocktail and for the 2H2 alone were nearly identical (Fig. 2f), with calculated IC50 values of 85 and 33 ng/mL, respectively. Considering that 2H2 constituted only 50% of antibodies in the cocktail, the neutralization data thus suggested that, in the presence of 3C1, 2H2 retains its neutralization potency. In another word, 2H2 is compatible with 3C1 for antibody cocktail formulation.

For future clinical application in humans, MAbs 3C1 and 2H2 were humanized by grafting the variable regions of murine antibodies (Supplementary Fig. 3b, c) onto human IgG1/kappa molecules (Supplementary Fig. 7a). The resulting human–mouse chimeric antibodies 3C1 (denoted as c3C1) and 2H2 (denoted as c2H2) were verified by western blotting with an anti-human IgG antibody (Supplementary Fig. 7b). Both c3C1 and c2H2

efficiently bound to recombinant SARS-CoV-2 RBD with $K_D$ values <1 pM, respectively, and to S trimer with $K_D$ values <1 pM, respectively (Supplementary Fig. 7c, e). In addition, both c3C1 and c2H2 could neutralize SARS-CoV-2 pseudovirus with calculated IC50 values of 0.758 and 0.054 µg/mL, respectively (Fig. 2g and Supplementary Fig. 7e). The neutralization potency of c3C1 and c2H2 was comparable to that of the corresponding mouse antibodies, indicating that human chimerization does not affect the functionality of 3C1 and 2H2. Moreover, the c3C1/c2H2 mixture (1:1 ratio) exhibited an IC50 of 0.054 µg/mL (Fig. 2g and Supplementary Fig. 7e), a neutralization potency similar to that of the murine MAb cocktail (Fig. 2f), demonstrating the compatibleness between c3C1 and c2H2. Furthermore, we tested the chimeric MAbs for neutralization of the SARS-CoV-2 pseudovirus carrying a D614G mutation in S protein, which currently predominates in the global pandemic and is implicated to have increased viral infectivity[33,34]. The results showed that c3C1, c2H2, and the c2H2/c3C1 cocktail remained highly neutralizing against the D614G pseudovirus (Supplementary Fig. 7f), suggesting a broad neutralization spectrum for the c2H2/c3C1 cocktail. At last, we assessed the chimeric MAbs for neutralization of authentic SARS-CoV-2 infection in vitro. The IC50 values for c3C1, c2H2, and the c2H2/c3C1 cocktail were determined to be 0.286, 0.022, 0.012 µg/mL, respectively (Fig. 2h and Supplementary Fig. 7e), suggesting that the combination of c3C1 and c2H2 has a synergistic effect on neutralization. Together, these results demonstrate that it is feasible to develop a humanized anti-SARS-CoV-2 antibody cocktail with therapeutic potential.

**MAbs 2H2 and 3C1 did not promote ADE in vitro.** A major concern in developing vaccines and therapeutic antibodies against coronaviruses is potential risk of antibody-dependent enhancement (ADE) that may exaggerate the disease[35,36]. Thus, our MAbs were evaluated for ADE potential by using two Fc receptor (FcR)-expressing cell lines as the target cell. Human THP-1 cells express both FcγRI and FcγRII[37], and K562 cells express human FcγRII[38]. Both THP-1 and K562 cells are capable of supporting mouse antibody-mediated enhancement of dengue virus infection[38–40]. We found that infection of either K562 or THP-1 cells with SARS-CoV-2 pseudovirus yielded only background levels of luciferase activity (~18 units), whereas the same amount of pseudovirus produced very strong luciferase signals averaging 915,470 units on 293T-hACE2 cells (Supplementary Fig. 8a, b), suggesting that SARS-CoV-2 pseudovirus entry into the two

FcγR-expressing cell lines was minimal. Moreover, treatment with serially diluted (ranging from 10 to 0.000128 μg/mL) 2H2 or 3C1 antibody did not significantly affect pseudovirus entry of the two cell lines (Supplementary Fig. 8a, b). Similarly, the humanized antibodies c2H2 and c3C1 did not show any enhancing effects on SARS-CoV-2 pseduovirus entry regardless of the antibody concentration (Supplementary Fig. 8a, b). Collectively, these results demonstrate that, in the assay system we tested, MAbs 2H2 and 3C1 did not promote ADE.

**In vivo prophylactic and therapeutic efficacies of the neutralizing MAbs**. To evaluate the protective efficacy of our MAbs, we developed in house a mouse model of authentic SARS-CoV-2 infection, in which wild-type Balb/c mice were intranasally inoculated with hACE2-encoded adenovirus 5 (Ad5-hACE2) to allow expression of the hACE2 receptor in the lung, followed by intranasal infection with live SARS-CoV-2 3 days later. This model permitted efficient SARS-CoV-2 infection and replication in the mouse lung; in contrast, only baseline levels of viral RNA were detected in the wild-type mice without Ad5-hACE2 inoculation after live virus challenge. Consistently, hematoxylin and eosin (H&E) staining assay showed that severe interstitial pneumonia was observed in the Ad5-hACE2-treated mice, but not in the mice without Ad5-hACE2 treatment (Fig. 3a). The prophylactic efficacy of MAb 2H2 was examined by intraperitoneally (i.p.) injecting 10 mg/kg (body weight) antibody into the Ad5-hACE2-treated mice 24 h before SARS-CoV-2 infection. Analysis of the viral load of the mouse lungs and histopathological examination at 3 days post infection (d.p.i.) showed that 2H2 pretreatment could almost completely neutralize SARS-CoV-2 infection, reducing viral load by ~1600 fold as compared to the control (PBS) pretreatment (Fig. 3a). To assess the therapeutic efficacy, Ad5-hACE2-treated mice were i.p. injected with 20 mg/kg of murine 2H2 antibody or 40 mg/kg of the c2H2/c3C1 (1:1, 20 mg/kg each) cocktail at 4 h.p.i., and mouse lungs were collected at 3 d.p.i. for qRT-PCR and H&E analysis. As shown in Fig. 3b, injection of 2H2 resulted in significant decrease (by ~17.8-folds) in viral load in the mouse lung and less severe lung lesions as compared to the control (PBS) treatment, indicating a strong therapeutic effect for 2H2; in addition, treatment with the c2H2/c3C1 cocktail appeared to be more effective than 2H2 alone in reducing viral loads in mouse lungs.

We also assessed the therapeutic potential of single c2H2 antibody and the c2H2/c3C1 cocktail administered at a delayed (24 h.p.i.) time point. Both c2H2 and the c2H2/c3C1 cocktail treatments could significantly reduce viral loads as compared to the control (PBS) treatment (Fig. 3b). Together, the above data demonstrate that the c2H2/c3C1 cocktail has high therapeutic efficacies in vivo.

**Structural snapshots of the S trimer in complex with 2H2 Fab**. To investigate the molecular basis of 2H2-mediated neutralization of SARS-CoV-2, we resolved four cryo-EM structures of the stabilized SARS-CoV-2 trimeric S glycoprotein in complex with 2H2 Fab in distinct conformational states, termed S-2H2-F1 (associated with one Fab), S-2H2-F2 (with two Fabs), and S-2H2-F3a/S-2H2-F3b (with three Fabs; Fig. 4a–d, Supplementary Fig. 9, and Supplementary Fig. 10a–f). Among these structures, S-2H2-F3a and S-2H2-F2 were better resolved to 3.8 and 4.3 Å resolution, respectively (Fig. 4a–d and Supplementary Fig. 10e–f). In the S-2H2-F1/S-2H2-F2 structures, there are one/two RBDs in the up configuration with each bound with a 2H2 Fab, while the remaining RBDs are in the down conformation without Fab binding (Fig. 4c, d and Supplementary Fig. 10c). Our S-2H2-F3a structure reveals that each of the three RBDs binds with a 2H2

Fab, with two RBDs in the up conformation (protomer 1 and 2), and the third RBD remaining down but can still engage with a 2H2 Fab (protomer 3, Fig. 4a, b). While in S-2H2-F3b structure, all the three RBDs are up and each binds with a Fab (Supplementary Fig. 10d).

These four cryo-EM structures with increasing number of associated Fabs may represent the snapshots during conformational transitions of the S trimer to gradually coordinate the binding of more 2H2 Fabs. In S-2H2-F1 state, we observed a further 9.2° outward tilt of the up RBD-1 induced by the first associated 2H2 Fab (Fig. 4e). Surprisingly, our S-2H2-F3a structure suggested although RBD-3 is in the down configuration, it can still bind a 2H2 Fab with a slight 3.8° upward tilt of RBD-3 and a further 12.4° outward tilt of RBD-2 to coordinately accommodate the binding of the third Fab (Fig. 4f). Collectively, this ensemble of cryo-EM structures revealed the conformational space of the S trimer as a dynamic allosteric machinery to coordinate the binding of more 2H2 Fabs.

Our structures show that 2H2 Fab is bound on the top of RBD (Fig. 4a–d). Inspection of the better resolved S-2H2-F3a structure revealed that the CDRH2 and CDRH3, together with all the three light chain complementarity-determining regions (CDRs) of 2H2 form contacts with the RBD domain, particularly the RBM region of the S protein (Fig. 4g and Supplementary Table 2), with the buried 2H2–RBD interaction surface area ranging from ~1320 to 1369 Å². The RBM also mediates the binding of S protein to human ACE2 (Fig. 4h), the receptor for both SARS-CoV-2 and SARS-CoV[1,14]. Among these interactions with RBD, the light chain of 2H2 Fab contributes more than the heavy chain does (Fig. 4i, j). Specifically, CDRL2 (residues 53–64), which touches the top "palm" of RBD, contributes the most to the interaction, possibly forming contacts with six residues (Y453, L455, loop[496–501], and Y505) in RBM. Both CDRL1 and CDRL3 contact the RBM loop[486–489] located on the other edge of the RBM, and CDRL1 also likely contacts A475 (Fig. 4i and Supplementary Table 2). As for the heavy chain, CDRH2 and CDRH3 mainly contact the RBM loop[483–490] (Fig. 4j and Supplementary Table 2). In addition to the CDRs, Q1 from the heavy chain of 2H2 Fab presumably interacts with V445 and G446 in RBM (Fig. 4j).

Furthermore, the epitope of 2H2 Fab on RBD would mostly overlap with the binding sites of ACE2 on RBD (13 out of the 17 total ACE2–RBD binding sites, PDB ID: 6M0J)[10], which could lead to severe clash and spatial hindrance between ACE2 and 2H2 Fab (Fig. 4h). Our structural data are in line with the observed high potency of 2H2 on blocking the interaction between RBD and ACE2 (Fig. 1c).

**Structural snapshots of the S trimer in complex with 3C1 Fab**. To disclose the molecular basis of 3C1-mediated neutralization of SARS-CoV-2, we determined four cryo-EM structures of SARS-CoV-2 S trimer in complex with the 3C1 Fab at distinct conformational states, including S-3C1-F1 (with one Fab), S-3C1-F2 (with two Fabs), and S-3C1-F3a/S-3C1-F3b (with three Fabs; Fig. 5a–d and Supplementary Fig. 11). Among these structures, S-3C1-F3b was better resolved to 4.3 Å resolution, and the other three were at 5.6–7.5 Å resolution range (Fig. 5a–d and Supplementary Fig. 12a–g). In the S-3C1-F3b structure, all the three RBDs are in the up conformation and each of them associates with a 3C1 Fab (Fig. 5a). Still, the S-3C1-F3b structure appears asymmetric especially in the associated RBD-3C1 Fab region. Indeed, all the three up RBDs in this structure exhibit an additional outward tilt relative to the up RBD in the open state S trimer likely induced by 3C1 binding, with RBD-1 outward tilt the most (50.8°, Fig. 5e). Due to the all up configuration of the

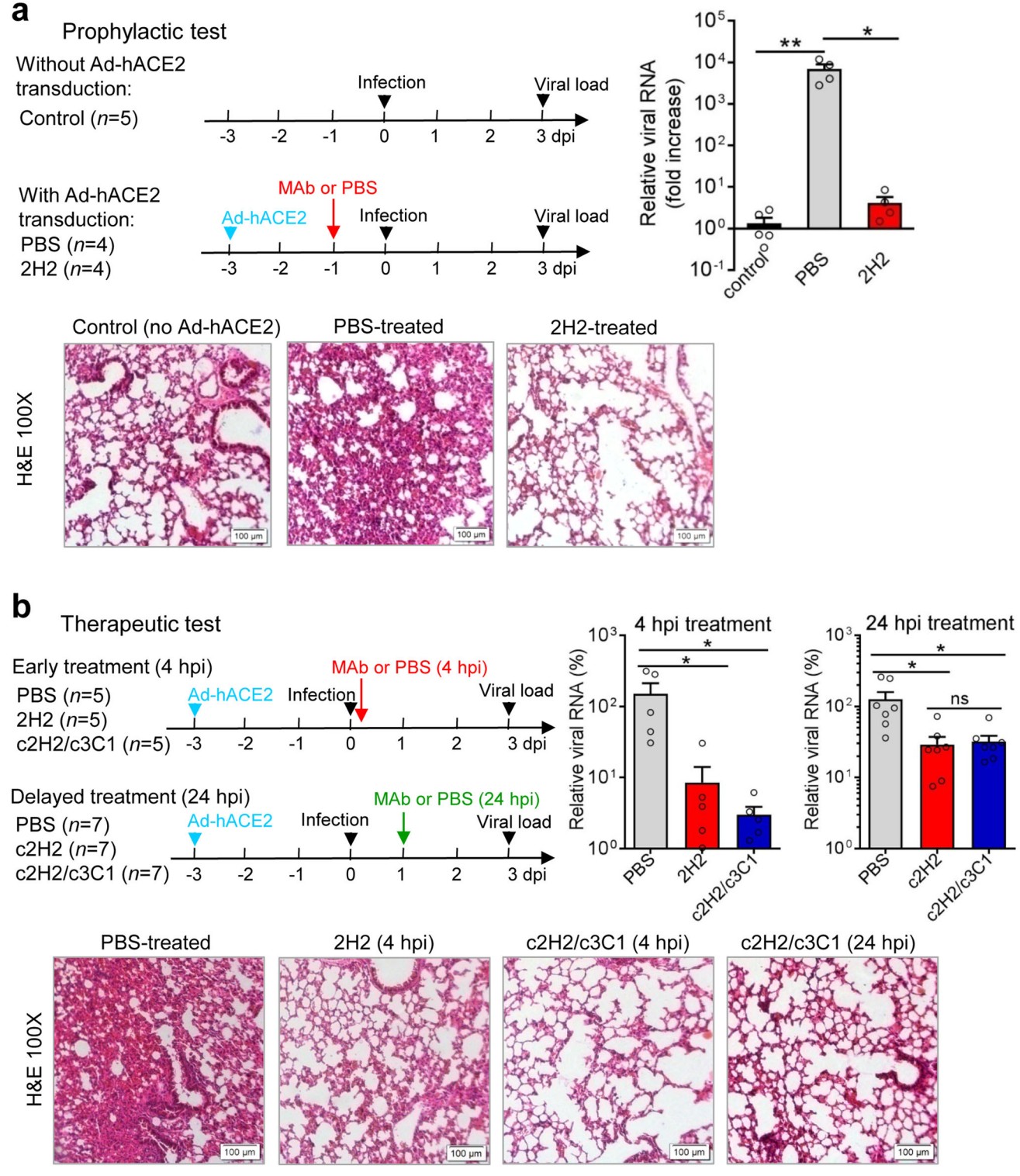

**Fig. 3 Protective efficacy of MAb 2H2 and the chimeric antibody cocktail against authentic SARS-CoV-2 infection in mice. a, b** In vivo prophylactic efficacy (**a**) and therapeutic efficacy (**b**) of MAb 2H2, c2H2, and/or the c2H2/c3C1 cocktail against SARS-CoV-2 infection. Upper left panel: study outline. Upper right panel: qRT-PCR analysis of viral RNA copies present in lung tissues after 3 days of infection. Lower panel: H&E staining of lung tissue sections at 3 d.p.i. For **a**, qPCR results are shown as fold increase relative to wide-type Balb/c group (without Ad5-hACE2 treatment). For **b**, qPCR results are expressed as viral RNA levels in different antibody treatment groups relative to that in the PBS control group. For top right panels in **a** and **b**, each symbol represents one mouse. Error bars represent SEM. Statistical significance was determined by a two-tailed Student's $t$ test and indicated as follows: ns not significant; *$p < 0.05$; **$p < 0.01$. For **a**, $p$ value between the control group and the PBS group (Ad-hACE2 transduction) is 0.0073; $p$ value between the PBS group and the 2H2 group is 0.0169. For early treatment experiment in **b**, $p$ value between the PBS group and the 2H2 group is 0.0488; $p$ value between the PBS group and the c2H2/c3C1 group is 0.0418. For delayed treatment experiment in **b**, $p$ value between the PBS group and the c2H2 group is 0.0183; $p$ value between the PBS group and the c2H2/c3C1 group is 0.0205; $p$ value between the c2H2 group and the c2H2/c3C1 group is 0.7803.

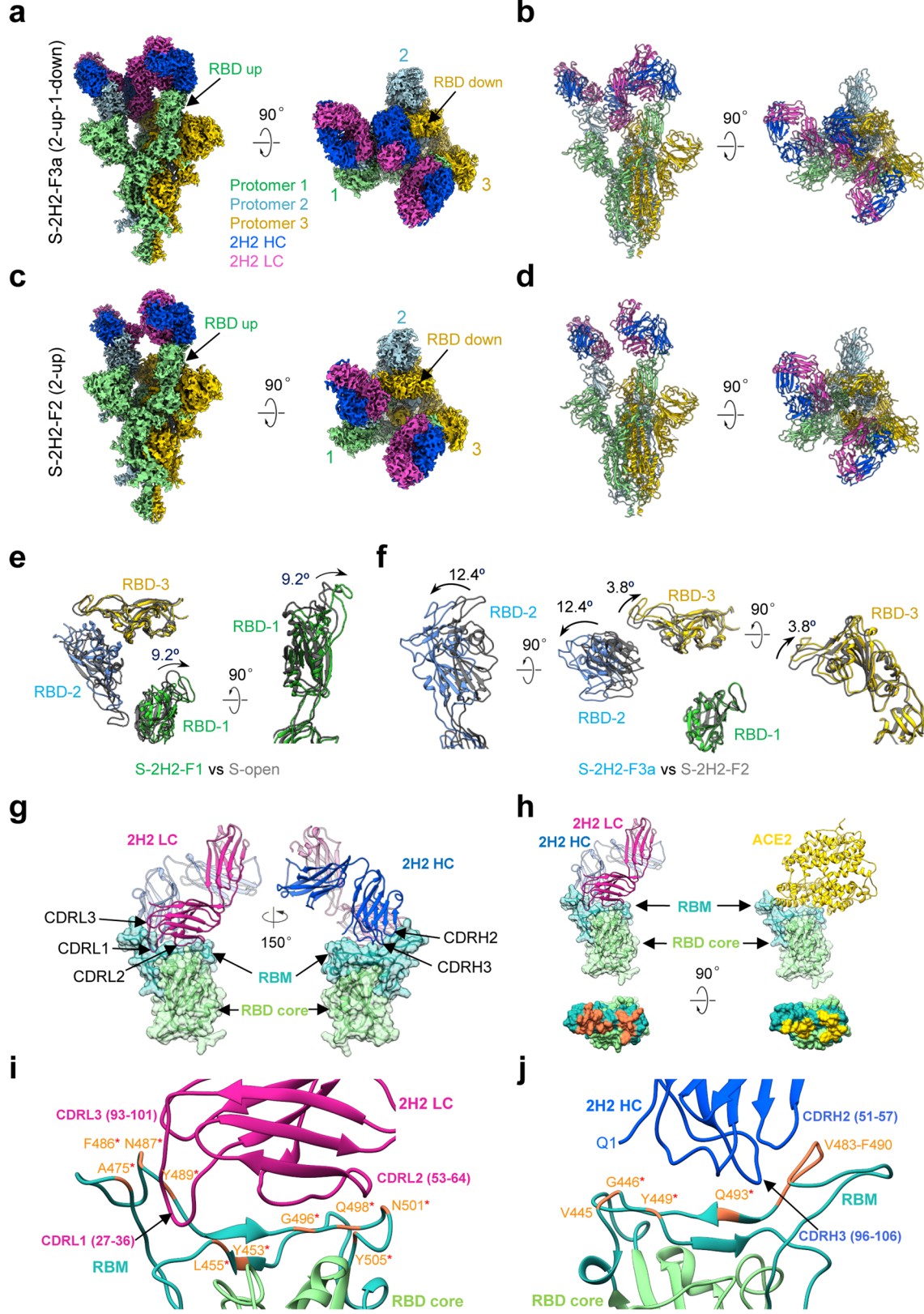

RBDs and their outward tilt, the originally covered S2 subunits are now exposed when visualized from the top (Fig. 5a, indicated with red circle), which might be beneficial for the release of S1 subunits and the subsequent transformations of the S trimer from the prefusion state to postfusion state. Furthermore, in the S-3C1-F3a map, there are two RBDs in the up configuration

(RBD-1 and RBD-2) and each binds a 3C1 Fab, while the remaining RBD-3 in the down conformation can still bind a 3C1 Fab (Fig. 5c). In addition, we also captured the S-3C1-F1/S-3C1-F2 structures with one/two RBDs up, and each of the up RBDs binds with a 3C1 Fab, while the remaining down RBDs have no associated Fab (Supplementary Fig. 12e, f).

**Fig. 4 Cryo-EM structures of the SARS-CoV-2 S trimer in complex with 2H2 Fab. a, b** Side and top views of the S-2H2-F3a cryo-EM map (**a**) and pseudo atomic model (**b**). RBD-1 and RBD-2 are in up configuration, while RBD-3 is down, with each of the RBDs bound with a 2H2 Fab. Protomer 1, 2, and 3 are shown in light green, powder blue, and gold, respectively. This color scheme is followed throughout. Heavy chain and light chain of 2H2 Fab in royal blue and violet red, respectively. **c, d** Side and top views of the S-2H2-F2 cryo-EM map (**c**) and pseudo atomic model (**d**), with two up RBDs (RBD-1 and RBD-2) each bound with a 2H2 Fab. **e, f** 2H2 Fab-induced conformational changes of the S trimer. Shown is the structural comparison of RBDs between S-2H2-F1 (in color) and S-open (dim gray) (**e**), and between S-2H2-F3a (in color) and S-2H2-F2 (dim gray) (**f**). **g** 2H2 Fab mainly binds to the RBM (light sea green surface) of RBD, with major involved structural elements labeled. RBD core is rendered as light green surface. **h** 2H2 Fab (left) and ACE2 (right, gold, PDB: 6M0J) share overlapping epitopes on RBM (second row) and would clash upon binding to the S trimer. **i, j** The involved regions/residues forming potential contacts between the light chain (in violent red, **i**) or heavy chain (in royal blue, **j**) of 2H2 and the RBD-1 of S-2H2-F3a. Asterisks highlight residues also involved in the interactions with ACE2. Note that considering the local resolution limitation in the RBD-2H2 portion of the map due to intrinsic dynamic nature in these regions, we analyzed the potential interactions that fulfill criteria of both < 4 Å side chain distance cutoff and <8 Å main chain distance cutoff, which criteria were followed throughout.

Interestingly, compared with the S-2H2 case, the four S-3C1 structures with increasing number of associated 3C1 Fabs reveal an overall similar pathway of the S trimer to gradually coordinate the binding of more Fabs (please see "Discussion" section). Our S-3C1-F1 structure also showed that the binding of the first 3C1 Fab could induce a further 32.3° outward tilt of the up RBD (Fig. 5f). Besides, our S-3C1-F3a structure revealed that to accommodate the third 3C1 Fab, RBD-3 in the down configuration exhibits a considerable upward tilt of 22.1° (Fig. 5g).

Unlike 2H2 bound on the top of RBD stretching out of the S trimer, 3C1 Fab is attached to the side of RBD (Fig. 5a–d). Inspection of the better resolved S-3C1-F3b structure revealed that the epitope involves mainly the β2-strand (T376 to C379) and loop[380–385] in the core region of RBD, and a small portion of RBM loop[501–506] (Fig. 5h and Supplementary Table 3). This observation is consistent with the biochemical data showing that substitution of the RBD core region of SARS-CoV-2 with the counterpart of SARS-CoV significantly reduced 3C1 binding to the resulting RBD mutant (Fig. 2d). As for the 3C1 Fab, all the three CDR loops of its light chain and also the heavy chain CDR1 and CDR2 loops contribute to the interaction (Fig. 5h). Further structural comparison of S-3C1-F3b with the SARS-CoV-2 RBD-ACE2 crystal structure (PDB ID: 6M0J)[10] suggested that ACE2 would clash with the heavy chain of 3C1 (Fig. 5i). Specifically, 3C1 heavy chain shares a small overlapping epitope on the RBM loop[502–505] region (including V503, G504, and Y505) with ACE2; additionally, the framework of 3C1-VH would clash with ACE2, which could be enhanced by the presence of an N-linked glycan at site N322 of ACE2 (Fig. 5i). This observation is in line with the ELISA result showing that 3C1 competes with ACE2 for binding to the RBD (Fig. 1c).

Surprisingly, although also bound to the side of RBD, 3C1 Fab can adopt a distinct orientation with 32.2° rotation when bound to the down RBD-3 (orientation 2) in S-3C1-F3a state, compared with that bound to the up RBD-3 (orientation 1) in S-3C1-F3b state, leading to slightly varied epitopes (Fig. 5j, k). These data also suggest the adaptability of 3C1 in coordinating the binding to RBD. Further structural analysis suggested that in orientation 1, only CDRH2 of 3C1 and CDRL2 of 2H2 share overlapping epitopes (residues 498–505 of RBM) and clash in nearby sites (Fig. 5l); while in orientation 2, there is no clash between 3C1 and 2H2 Fabs, allowing simultaneous binding of RBD by 2H2 and 3C1 (Fig. 5m). Our antibody binding competition assay had shown that 2H2 and 3C1 could noncompetitively bind to RBD (Fig. 2a, b). Therefore, both 2H2 and 3C1 may simultaneously bind to RBD through adapting feasible orientation of 3C1 or coordinated conformational changes, avoiding potential spatial conflicts.

Moreover, in the same cryo-EM dataset of S-3C1, we additionally resolved a ligand-free close and an open S trimer structure, termed S-closed and S-open at the resolution of 3.0 and

6.3 Å, respectively (Supplementary Figs. 11 and 12c, d). The S-closed structure resembles the tightly closed ground prefusion state captured in our recent study[14]. Of note, compared with the S-closed state, the S-3C1-F1/S-2H2-F1 structures both exhibit an untwisting of the S1 subunit region (Supplementary Figs. 10g and 12i), with the fusion peptide (FP) released. The conformational landscape distribution suggested that ~53.0 % of the particles in the S-3C1 dataset has the 3C1 Fab engaged (Fig. 6b), a ratio much lower than that observed in the S-2H2 dataset (~100 %). This population difference may reflect the relatively weaker RBD binding ability of 3C1 compared with that of 2H2 (Fig. 1a, b).

## Discussion

In the present study, we discovered and comprehensively characterized two groups of potent anti-SARS-CoV-2-neutralizing MAbs that target distinct epitopes on RBD. MAb 2H2, the representative of the antibody group whose epitope largely overlaps the ACE2 RBM on RBD, was potent in neutralizing authentic SARS-CoV-2 infection in vitro with IC50 of 0.007 µg/mL (Fig. 1e), and in conferring protection in vivo both prophylactically and therapeutically (Fig. 3). MAb 3C1, the sole member of another antibody group, mainly binds the side (specifically the T376 to C379 β-strand) of the RBD core and defines a previously unreported neutralizing antibody epitope. Although 3C1 is a relatively weaker neutralizer as compared to 2H2, it is compatible with 2H2 for formulating a noncompeting antibody cocktail. Such an antibody cocktail will likely be able to prevent rapid mutational escape seen with individual antibodies as reported recently[30]. We also demonstrated chimeric MAbs c2H2 and c3C1 retain binding and inhibitory functions toward SARS-CoV-2 (Fig. 2 and Supplementary Fig. 7) and the c2H2/c3C1 cocktail could effectively treat authentic SARS-CoV-2 infection in the mouse model (Fig. 3). Particularly, injection of SARS-CoV-2-infected mice with the c2H2/c3C1 antibody cocktail at 24 h.p.i. could significantly reduce viral replication in the mouse lungs (Fig. 3), demonstrating that the delayed antibody treatment remains therapeutic effective. We should mention that, in the recently published works evaluating the therapeutic efficacies of anti-SARS-CoV-2-neutralizing MAbs in mouse models, all antibody treatments were initiated no later than 12 h.p.i. (refs. [20,29]). Because in real clinical settings, the patients are often diagnosed with SARS-CoV-2 several days to 2 weeks after contracting the virus and therefore an therapy are always initiated at a relatively delayed time point, the extended therapeutic window (up to 24 h.p.i.) of our MAb cocktail shown in this study, thus highlights its advantage and strong potential as a therapeutic drug candidate worthy of further development.

ADE is an important safety issue that needs to be addressed during the development of vaccines and therapeutic antibodies against coronaviruses[35,36]. A recent report showed that neutralizing

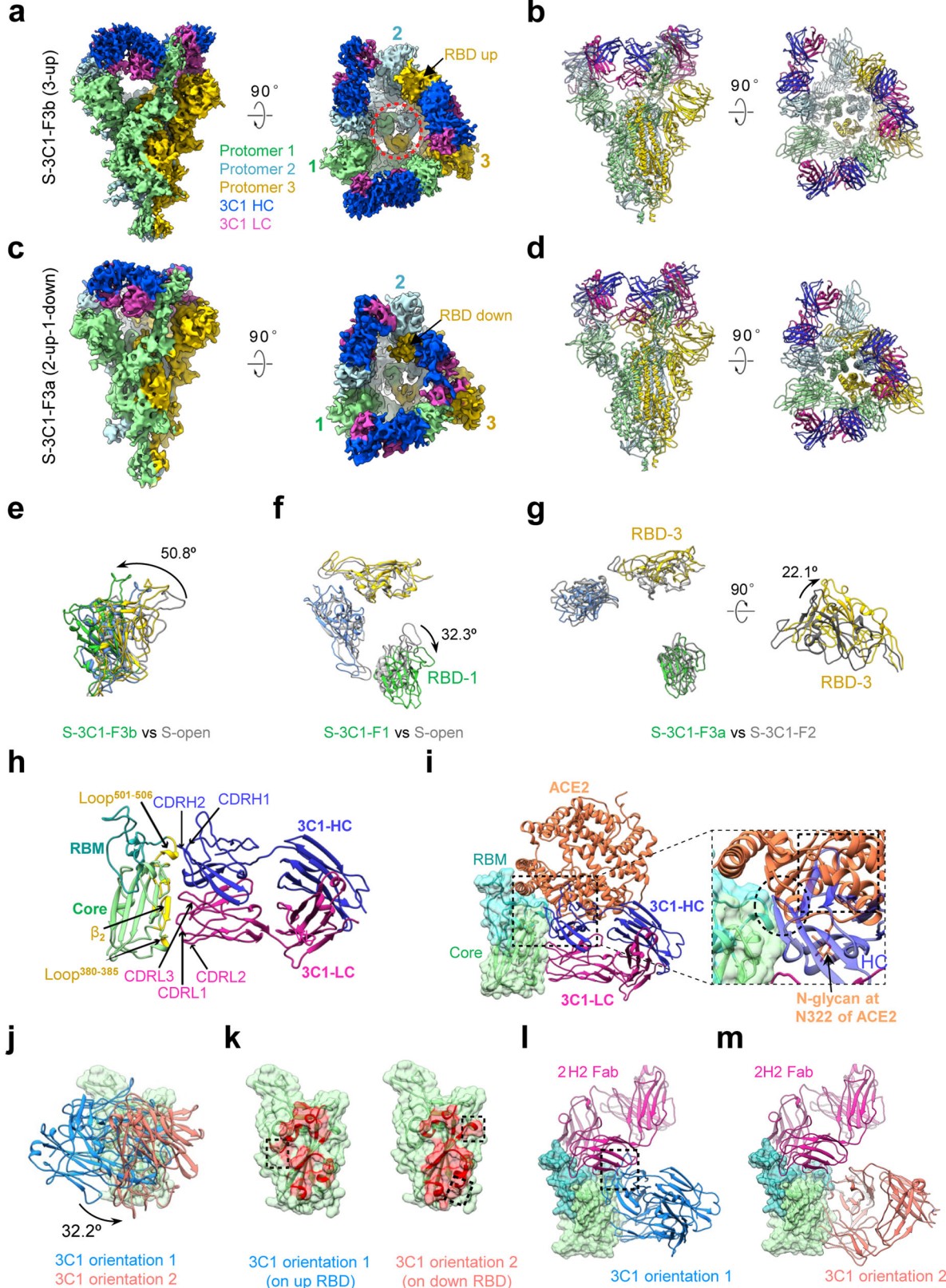

MAbs targeting the RBD of either MERS-CoV or SARS-CoV could enhance pseudovirus entry into FcR-expressing cell lines, including THP-1 (ref. [41]). The ADE assay described in that study was modified and then used in the present study to evaluate the ADE potential of our anti-SARS-CoV-2 MAbs. We found that neither the murine antibodies nor the chimeric antibodies with human

IgG1 Fc could enhance SARS2-CoV-2 pseudovirus entry of FcR-expressing K562 or THP-1 cells regardless of the antibody concentration (Supplementary Fig. 8). We should mention that the same cell lines have been shown to support anti-DENV-E antibody-triggered ADE of DENV infection in previous studies[38–40]. Our data appear contradictory to the results from previous studies on

**Fig. 5 Cryo-EM structures of the SARS-CoV-2 S trimer in complex with the 3C1 Fab. a, b** S-3C1-F3b cryo-EM map (**a**) and pseudo atomic model (**b**). All the three RBDs are up and each of them binds with a 3C1 Fab. The heavy chain of the 3C1 Fab in medium blue and light chain in violet red. **c, d** S-3C1-F3a cryo-EM map (**c**) and pseudo atomic model (**d**). There are two up RBDs and one down RBD, with each bound with a 3C1 Fab. **e** Structural alignment of the three up RBDs of S-3C1-F3b (in color) and the only up RBD from S-open (gray), suggesting 3C1 induced outward tilt of the RBDs within the S trimer. **f, g** Conformational comparison between S-3C1-F1 and S-open (**f**), as well as between S-3C1-F3a and S-3C1-F2 (**g**). **h** RBD/3C1 interaction interface (take RBD-3/3C1 of S-3C1-F3b as an example), with major involved structural elements labeled. **i** ACE2 (coral, PDB: 6M0J) would clash with the heavy chain of 3C1 Fab (blue). They share overlapping epitopes on the RBM (dotted black circle); additionally, the framework of 3C1-VH would clash with ACE2 (dotted black frame), which could be enhanced by the presence of an N-linked glycan at site N322 of ACE2. **j** 3C1 showed two distinct orientations to bind RBD within S trimer, i.e., adopting orientation 1 to associate with up RBD while orientation 2 with down RBD. **k** Contact footprint variations of 3C1 on up RBD (left) compared with that on down RBD (right), with unique epitopes indicated by dotted black frame. **l–m** Potential simultaneous binding of RBD by 2H2 and 3C1 cocktail. In 3C1 orientation 1, 3C1 and 2H2 could have minor clash (indicated by black frame, **l**); while in origination 2, there is no clash between 3C1 and 2H2 Fabs (**m**).

the anti-MERS-CoV or anti-SARS-CoV MAbs (Wan et al.[41]). Although the mechanism underlying such a contradiction remains to be elucidated, we speculate that SARS-CoV-2 may require an unidentified host factor for MAb-bound pseduovirus or virion to enter FcR-bearing cells, and such a host factor is lacking in the K562 and THP-1 cells used in the present study, whereas MERS-CoV and SARS-CoV may not need the assistance of the same host factor for entry of FcR-expressing cells. It is also possible that MAb-induced drastic conformational changes of SARS-CoV-2 S proteins (as shown in Fig. 6) may cause premature shedding of S1 and exposure of the FP before the virion reach the cell surface, thereby abolishing viral infectivity. Nonetheless, our study demonstrated that MAbs 2H2 and 3C1 do not promote ADE in vitro at least not in the assay system we used. In addition, our mouse challenge experiments showed that the 2H2 antibody or the c2H2/c3C1 cocktail were able to neutralize, but not enhance SARS-CoV-2 infection in vivo (Fig. 3). Together, these data demonstrate proof-of-concept for the application of our MAbs as a safe and effective treatment option against SARS-CoV-2 infection. We should point out that c2H2 and c3C1 are mouse–human chimeras, and therefore for future human use the antibodies will need to be further humanized by grafting their CDRs into a suitable human MAb backbone.

We have recently showed that in the ligand-free condition, SARS-CoV-2 S trimer dominantly adopts a stable tightly closed ground prefusion conformation (no RBD up), with only a minor population of the particles in the open conformation with one RBD up[14]. Here, our structural data showed that binding of 3C1 or 2H2 MAbs could trigger dramatic conformational transitions of the S trimer (especially in the S1 subunit region, Supplementary Fig. 12h) from the tightly closed ground prefusion state to the unstable, loosely packed open state with two or three RBDs up and released FP. In another word, the prefusion S protein is destabilized to some extent by the 2H2/3C1 Fabs. Note that the SARS-CoV-2 S protein used in this study was stabilized (tentatively in the prefusion conformation) by furin cleavage site mutation and two consecutive proline mutations[13,14]. Therefore, it is very likely that our Fabs could induce even further conformational changes toward the postfusion state in the wild-type S protein. Upon ACE2 receptor binding, SARS-CoV S protein undergoes similar conformational changes with more than one RBD up and to transit to the postfusion state[42]. As for SARS-CoV-2, we recently also observed ACE2 receptor-triggered transitions of S trimer toward fusion-prone or postfusion states[14]; and combined with our current results, it appears the untwisting of S1 induced by ACE2 receptor/MAb binding could release the originally packed FP and induce an early exposure of FP. Therefore, MAb-induced transition of S protein from prefusion to fusion or postfusion states, accompanied by premature release of S1, and exposure of the cleavage site and FP, may disrupt the integrity of the virion and render the virus defective. Hence, our study suggests that, besides blockade of the interaction between the virus

and ACE2, destabilization of the virion is possibly another neutralization mechanism for our MAbs.

In this study, we captured four distinct structural states for each of the S protein/Fab complex (Figs. 4–6), allowing us to glimpse the main features of the dynamic process of conformational transitions induced by Fab binding, i.e., from binding with one Fab to with three Fabs making all the RBDs occupied. We should emphasize that the Fab-induced conformational changes in S trimer cannot be observed in the RBD/Fab crystal structures, highlighting the advantage of cryo-EM in capturing dynamic conformational shifts in macromolecular complex-Fab recognition. It has been hypothesized that for SARS or MERS, the up conformation (active state) of the RBD is required for the binding of neutralizing MAbs directed at the receptor-binding site[43,44]. However, here for both Fabs, we observed a state (S-2H2-F3a or S-3C1-F3a) with RBD-3 in the down conformation, but can still associate with a Fab. This suggests that for the antibody targeting RBD domain, if only the epitope is exposed and there is enough space to accommodate the Fab, the RBD can be "grasped" by the Fab regardless it is in the up or down conformation.

Altogether, we propose a model of stepwise binding of 2H2/3C1 Fabs to the RBD domain of the SARS-CoV-2 S trimer (Fig. 6). Take 2H2 as an example, in step 1, the only up RBD in the S-open state binds one 2H2 Fab first, leading to the S-2H2-F1 state. Our recent study suggested that in the S-open state, the RBD-2 adjacent to the up RBD-1 has encoded intrinsic dynamics[14]. Thus, once RBD-1 bound with a Fab, the resulting outward tilt of RBD-1 (Figs. 4e and 5f) could break the allosteric constrains originally imposed on RBD-2, leading to an up configuration of RBD-2. This is in line with a recent study of S trimer on intact SARS-CoV-2 virion, showing that there is a minor population of the S trimer with the RBD-2 also in the up conformation[45]. In step 2, the transiently up RBD-2 can be quickly trapped by 2H2 Fab, resulting in the S-2H2-F2 form with each of the two up RBDs bound with a Fab. Consequently, the original steric hindrance could be released to allow RBD-3 to expose its buried epitope and also leave enough space to accommodate the binding of the third 2H2 Fab. Indeed, we observed a further 12.4° outward tilt of RBD-2, releasing the space for the third Fab (Fig. 4f). Interestingly, in step 3, our data suggested that there are two possible reaction pathways. In pathway one, the down RBD-3 with exposed epitope can now bind a Fab, forming S-2H2-F3a with each of the RBDs (two up and one down) bound with a Fab. In pathway two, after the RBD-1 and RBD-2 are all up releasing the original allosteric constrains, RBD-3 has more chance to be transiently up, which could be encoded in the S trimer conformational space or triggered by certain external factors. This up RBD-3 can then be trapped by a Fab and retained in the up conformation, forming S-2H2-F3b with each of the three up RBDs bound with a Fab. The two pathways in step 3 could take place simultaneously.

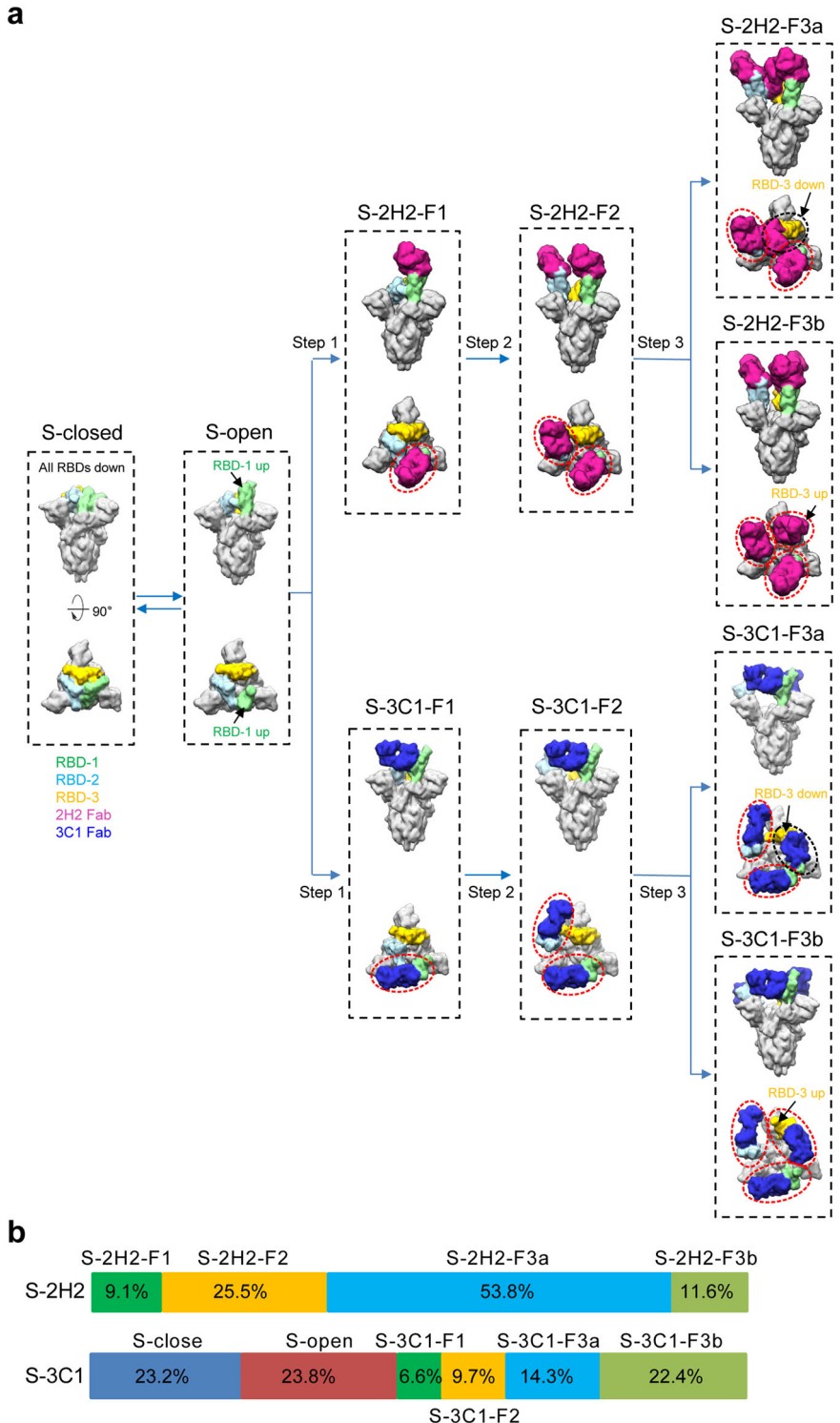

**Fig. 6 A proposed model of stepwise binding of 2H2/3C1 Fabs to the RBD of SARS-CoV-2 S trimer. a** 2H2 and 3C1 Fabs appear to follow similar pathway to induce generally comparable conformational transitions of the S trimer to neutralize the virus. RBD-1, RBD-2, and RBD-3 are colored in light green, light blue, and gold, respectively; 2H2 and 3C1 Fab in violent red and medium blue, respectively. Red ellipsoid and black ellipsoid indicate Fab bound to up RBD and down RBD, respectively. The maps of S-2H2 and S-3C1 complexes shown here were generated by lowpass filtering of the corresponding models to 10 Å resolution. **b** Population distribution for the S-2H2 and S-3C1 dataset.

Strikingly, our structural data further suggest that for the 3C1 Fab, although its epitope mostly locates on the side of RBD, distinct from that of 2H2, it appears to follow similar pathway to induce generally comparable conformational transitions of the S trimer. We therefore postulate that this procedure might also be adopted by other MAbs targeting similar regions of RBD. Thus, our structural study reveals that binding of neutralizing MAbs to SARS-CoV-2 S trimer is a well-coordinated dynamic process involving stepwise allosteric conformational changes of the S trimer, and also sheds light on the structural basis for MAbs 2H2

and 3C1 as noncompeting antibody cocktail. These structural information enhances our understanding of anti-SARS-CoV-2 antibody-mediated neutralization and protection.

## Methods

**Cells and viruses**. SP2/0 mouse myeloma cells were grown in RPMI 1640 medium (Gibco, Thermo Fisher, USA) supplemented with 10% fetal bovine serum (FBS; Gibco) at 37 °C. African green monkey kidney VeroE6 cells were cultured in DMEM (Gibco, USA) supplemented with 10% FBS. HEK 293 F suspension cells (Thermo Fisher) were grown in FreeStyle 293 expression medium (Gibco). Expi-CHO-S™ cells (Thermo Fisher) were grown in ExpiCHO expression medium (Gibco). SARS-CoV-2 clinical isolate nCoV-SH01 (GenBank: MT121215.1)[46] was expanded in VeroE6 cells and virus titers were expressed as plaque forming units (PFU) per mL. All the infection experiments were performed in the biosafety level-3 (BSL-3) laboratory of Fudan University.

**Recombinant proteins and antibodies**. For mouse immunization, recombinant SARS-CoV-2 RBD (residues R319 to F541) fused with a C-terminal mouse IgG1 Fc tag (RBD-mFc) was purchased from Sino Biological Inc (Beijing, China). For antibody screening and characterization, several recombinant proteins were produced in our laboratory. Specially, to prepare SARS-CoV-2 RBD, RBD DNA fragment corresponding to residues V320 to G550 derived from SARS-CoV-2 strain Wuhan-Hu-1 (GenBank ID: MN908947.3) was codon optimized and cloned into a modified pcDNA3.4 vector that contains interleukin-10 (IL-10) signal sequence and a C-terminal His tag, yielding plasmid pcDNA3.4-SARS-2 RBD. To express SARS-CoV RBD, RBD gene fragment corresponding to rsidues R306 to I520 derived from SARS-CoV strain Tor2 (GenBank ID: AAP41037.1) was codon optimized and cloned into the expression secretion vector pSecTag2A (Invitrogen, USA), yielding plasmid pSecTag2A-SARS-RBD. To generate ACE2, DNA fragment encoding the extracellular domain of human ACE2 (residues Q18 to S740) was cloned into a modified pcDNA3.4 vector that contains IL-10 signal sequence and C-terminal human IgG1 Fc and His tag, yielding plasmid pcDNA3.4-ACE2-hFc. To prepare SARS-CoV-2 S protein, mammalian codon-optimized gene coding S ectodomain (residues 1–1208) with proline substitutions at residues 986 and 987, a "GSAS" substitution at the furin cleavage site (residues 682–685) was cloned into vector pcDNA3.1+. A C-terminal T4 fibritin trimerization motif, a TEV protease cleavage site, a FLAG tag and a His tag were cloned downstream of the SARS-CoV-2 S glycoprotein ectodomain. Primer information is listed in Supplementary Table 4. The above four plasmids were separately transfected into HEK 293 F suspension cells using polyethylenimine (PEI; PolySciences, USA). The supernatants were harvested after 4–5 days of culture and His-tagged proteins were purified using Ni-NTA resin (Millipore, USA) according to manufacturer's protocol. To prepare biotinylated proteins, purified ACE2-hFc fusion protein or SARS-CoV-2 S trimer protein were dialyzed against PBS and then labeled with EZ-Link™ Sulfo-NHS-LC-LC-Biotin (Thermo Fisher) followed by purification using Zeba™ spin desalting column (Thermo Fisher), according to manufacturer's instructions.

MAb 5F8 is an IgG1 antibody against E protein of zika virus[47], serving as isotype control.

**Preparation of MAbs**. The animal studies were approved by the Institutional Animal Care and Use Committee at the Institut Pasteur of Shanghai. The mice were kept in the SPF (specific pathogen free) animal facility with controlled temperature (20–26 °C), humidity (40–70%), and lighting conditions (12 h light/12 h dark cycle).

To generate MAbs, female BALB/c mice aged 6–8 weeks were each primed with 100 µg of RBD-mFc protein (Sino Biological) formulated with 0.5 mg of aluminum hydroxide adjuvant (Invivogen, USA) and 25 µg of CpG oligonucleotides (Sangon Biotech, China) via the i.p. route on day 0. The mice were boosted via the subcutaneous route on day 8 with RBD-mFc (50 µg/mouse) emulsified with Freund's complete adjuvant (Sigma), and on day 13 with RBD-mFc (50 µg/mouse) emulsified with Titermax adjuvant (Sigma). On day 22, one mouse was injected with 75 µg of HEK 293F-expressed RBD protein in PBS in a tail vein. On day 26, splenocytes were isolated and fused with SP2/0 cells using polyethylene glycol 1450 (Sigma). Fused cells were then selected in a hypoxanthine, aminopterin, and thymidine (HAT; Sigma) medium. Eight days later, hybridoma supernatants were screened for their ability to bind to RBD protein and to block the ACE2-hFc/SARS-CoV-2 RBD binding by ELISA, as described below. ELISA-positive hybridoma cells were cloned by limiting dilution method and the resulting monoclonal cell lines were expanded. Purified MAbs were prepared from ascitic fluids using HiTrap™ Protein G HP column (GE Healthcare, USA).

**ELISA**. To determine binding properties of the antibodies, ELISA plates (Nunc, USA) were coated with 100 ng/well of HEK 293F-expressed SARS-CoV-2 RBD or SARS-CoV RBD (purchased from Kactus Biosystems (Shanghai, China) or produced in the laboratory) at 4 °C overnight. The plates were then blocked with 5% milk in PBS-Tween 20 (PBST). After washing with PBST, 50 µL of hybridoma culture supernatants or serially diluted purified MAbs were added to the wells and incubated at 37 °C for 2 h. After washing, horseradish peroxidase (HRP)-conjugated anti-mouse IgG (Sigma; diluted 1:10,000 in 1% milk/PBST) was added and incubated at 37 °C for 1 h. After washes and color development, absorbance was monitored at 450 nm.

For receptor competition assay, microplates (Nunc) were coated at 4 °C overnight with 40 ng/well of HEK 293F-expressed SARS-CoV-2 RBD and then blocked with 5% milk/PBST. After washing with PBST, 25 µL of hybridoma culture supernatants or serially diluted purified MAbs were mixed with 25 µL (20 ng) of unlabeled or biotinylated ACE2-hFc, and the mixtures were added to the wells and incubated at 37 °C for 2 h. The corresponding secondary antibodies, HRP-conjugated anti-human IgG (Abcam, USA; diluted 1:8,000 in 1% milk/PBST) or HRP-conjugated streptavidin (Life Technologies, USA), were added and incubated at 37 °C for 1 h. After washes and color development, absorbance was monitored at 450 nm.

To determine the isotypes of the MAbs, hybridoma culture supernatants were tested by sandwich ELISA using the SBA Clonotyping system/HRP kit (Southern Biotech, USA) according to manufacturer's instructions.

**Pseudovirus neutralization assay**. Murine leukemia virus (MLV)-based SARS-CoV-2 S pseudoviruses were prepared as follows: HEK 293 T cells grown in 10-cm dish were cotransfected using PEI (polysciences) with 10 µg of MLV Gag-Pol packaging plasmid, 10 µg of transfer plasmid containing a luciferase or EGFP reporter gene, and 2 µg of plasmids encoding either wild-type or mutant (D614G) S proteins. The cells were incubated with the transfection mixture for 4 h. After washing once with DMEM, fresh DMEM medium supplemented with 10% FBS was added and incubated at 37 °C for 48 h. The culture supernatant was harvested, filtered through 0.45 µm filters and either used immediately or frozen at −80 °C.

For pseudovirus neutralization assay, VeroE6 cells or HEK 293 T cells stably overexpressing human ACE2 receptor were plated into 96-well or 48-well plates and grown overnight. A total of 90 µL of the pseudovirus was mixed with 45 µL of antibody samples (hybridoma culture supernatants or serially diluted purified MAbs), and the mixtures were incubated at 37 °C for 1 h, and then added to the plates. After 2 h, the pseudovirus/antibody mixtures were removed and the cells were washed once with DMEM, followed by the addition of fresh culture medium. At 48 h after infection, luciferase activity was measured using the luciferase assay system (Promega), or GFP expression resulting from pseudovirus infection was analyzed by flow cytometry using a FACSCelesta flow cytometer (BD Biosciences, USA).

**Determination and analysis of MAb sequences**. For antibody sequencing, total RNA was extracted from hybridoma cells and first strand cDNA is prepared using M-MLV reverse transcriptase (Promega) and MAb isotype-specific primers. DNA fragments encoding antibody variable regions were amplified individually from cDNA using mouse Ig-primer set (Novagen, Merck, Germany) and Premix Ex Taq reagent (Takara, Japan), followed by DNA sequencing.

The closest mouse immunoglobulin V, D, and J germline genes and positions of CDRs were identified using IgBLAST[32].

**Bio-layer interferometry assay**. To measure binding affinities of the MAbs, BLI experiments were performed using an Octet Red96 instrument (Pall FortéBio, USA) following manufacturer's instructions. Briefly, in one experiment, His-tagged RBD proteins of SARS-CoV-2 or SARS-CoV were immobilized to Ni-NTA biosensors (Pall FortéBio) until saturation. In another experiment, biotinylated SARS-CoV-2 S trimer protein was immobilized to streptavidin (SA) biosensors (Pall FortéBio) until saturation. For both experiments, the antigen-immobilized biosensors were transferred to wells containing MAb samples at varying concentrations for a 500-s association step. The sensors were then transferred to dissociation buffer (0.01 M PBS supplemented with 0.02% Tween 20 and 0.1% bovine serum albumin) for a 500-s dissociation step.

For antibody competition assay, the antigen-immobilized biosensors were then dipped into the wells containing 15 µg/mL (100 nM) of the first MAb for a 500-s association period. The sensors were then transferred to wells containing dissociation buffer or 15 µg/mL of the second MAb samples and incubated for 500 s. For all BLI assays, data analysis was performed using Octet data analysis software version 11.0 (Pall FortéBio).

**Authentic virus neutralization assay**. A total of 200 PFU (50 µL) of live SARS-CoV-2 virus (nCoV-SH01 strain) was mixed with 50 µL of fourfold serially diluted purified MAbs and incubated at 37 °C for 1 h. The mixtures were then added to confluent VeroE6 cells grown in 96-well plates. After 48 h of incubation at 37 °C, culture supernatants were harvested for viral RNA isolation and cells were fixed for immunofluorescence analysis.

RNA was extracted from culture supernatants using TRIzol reagent (Invitrogen, USA). Reverse transcription quantitative PCR (RT-qPCR) was carried out in an MXP3000 thermal cycler (Stratagene, USA) using Verso SYBR Green 1-Step qRT-PCR Kit Plus ROX Vial (Thermo Fisher) according to the manufacturer's protocol. The primers that target SARS-CoV-2 N gene spanning nt 608–706 are as follows: forward primer, 5′-GGGGAACTTCTCCTGCTAGAAT-3′; reverse primer, 5′-CAGACATTTTGCTCTCAAGCTG-3′.

For immunofluorescence assays, cells were fixed in 4% paraformaldehyde solution, and permeabilized with 0.2% Triton X-100 (Thermo Fisher). Next, the cells were incubated overnight at 4 °C with a mouse polyclonal antibody against N protein prepared in house, followed by incubation with Alexa Fluor 488-labeled donkey anti-mouse IgG secondary antibody (1:1000, Thermo Fisher) at 37 °C for 1 h. Cell nuclei were stained with DAPI (Thermo Fisher). Finally, the images were recorded by fluorescence microscopy (Thermo Fisher).

**Mapping of MAb epitopes with RBD mutants by ELISA.** For antibody epitope mapping, a series of SARS-CoV-2 RBD mutants were constructed. In all RBD mutants, residues from the core or RBM regions of SARS-CoV-2 RBD were replaced with the corresponding residues of SARS-CoV to make the chimeric proteins (Supplementary Fig. 6). Specially, for mutant RBD (Core), residues R319 to N437 in the core region were mutated; for mutant RBD (RBM-R1), residues S438 to G446 in the RBM region were substituted; for mutant RBD (RBM-R2), residues L452 to K462 in the RBM region were replaced; for mutant RBD (RBM-R3), residues T470 to T478 in the RBM region were substituted; for mutant RBD (RBM-R4), residues N481 to F486 in the RBM region were mutated; for mutant RBD (RBM-R5), residues F490 to V503 in the RBM region were replaced. All mutant plasmids were constructed based on the plasmid pcDNA3.4-SARS-2 RBD by using the Mut Express$^{TM}$ II Fast Mutagenesis Kit V2 (Vazyme, China) according to the manufacturer's protocol. The resulting mutated plasmids were separately transfected into HEK 293 F cells using PEI. After 5 days of culture, His-tagged proteins were purified from the culture supernatants using Ni-NTA resin (Millipore).

The RBD mutants were tested for reactivity with the MAbs by ELISA. Briefly, microplates were coated at 4 °C overnight with 100 ng/well of individual RBD mutant in PBS. After blocking, the plates were incubated with the mAbs (50 ng/well) at 37 °C for 2 h, followed by incubation with HRP-conjugated anti-mouse IgG (Sigma; diluted 1:10,000 in 1% milk/PBST). After color development, absorbance at 450 nm was determined.

**Generation and characterization of chimeric MAbs.** To prepare chimeric MAbs, DNA fragments encoding variable regions of murine MAbs were cloned into modified pcDNA3.4 vectors that contain IL-10 signal sequence and the constant regions of human IgG1 or kappa chains by using ClonExpress II One Step Cloning Kit (Vazyme, China). Heavy chain and light chain expression plasmids were transiently cotransfected in ExpiCHO-S™ cells (Thermo Fisher) by using the ExpiFectamine CHO transfection kit (Gibco). The supernatants were harvested after 14 days of culture and the MAbs were purified using protein G agarose resin 4FF (Yeasen, China), according to manufacturer's protocol.

To characterize chimeric MAbs, the recombinant chimeric MAbs were subjected to BLI and pseudovirus neutralization assays as described above.

**ADE assay.** A total of 150 μL of the SARS-CoV-2 pseudovirus was mixed with 50 μL of fivefold serially diluted antibody samples, and the mixtures were incubated at 37 °C for 2 h. Next, 30,000 THP-1 or K562 cells were plated into 48-well plates, followed by addition of the pseudovirus/antibody mixtures. Three days after infection at 37 °C, the cells were transferred to 1.5-mL Eppendorf tubes and and washed once with PBS. Luciferase activity was measured using the luciferase assay system (Promega).

**In vivo protection assays.** To generate recombinant adenovirus 5 expressing human ACE2 (Ad5-hACE2), hACE2 gene fragment was cloned into the shuttle vector pShuttle-CMV[48], resulting in plasmid pShuttle-CMV-hACE2. This plasmid was linearized by PmeI digestion and then used to transform BJ5183-AD-1 cells (Weidi, China), resulting in plasmid pAd5-hACE2. Next, the pAd5-hACE2 plasmid was linearized with PacI and transfected in HEK 293 cells, to rescue adenovirus Ad5-hACE2. Ad5-hACE2 was amplified on HEK 293 cells and purified by CsCl gradient centrifugation. Adenovirus titer was determined using OD260 assay and the titer (virus particles [VP]/mL) can be calculated by multiplying the OD260 reading by $1.1 \times 10^{12}$.

Wild-type male Balb/c mice (6–8 week) were raised in pathogen-free isolation cages in the BSL-3 laboratory of Fudan University and received humane care in compliance with the guidelines of the Animal Research Ethics Board of Fudan University. Mice were transduced intranasally with $5 \times 10^{10}$ VP of Ad5-hACE2 at −3 d.p.i. To assess the prophylactic efficacy of MAbs, groups of Ad5-hACE2-transduced mice were injected i.p. with PBS or 10 mg/kg of MAb 2H2 at −1 d.p.i. and then infected intranasally with $2 \times 10^5$ PFU of SARS-CoV-2. To evaluate the therapeutic efficacy of MAb treatment, groups of Ad5-hACE2-transduced mice were infected intranasally with $2 \times 10^5$ PFU of SARS-CoV-2. The infected mice were injected i.p. with PBS, 20 mg/kg of MAb 2H2, or the chimeric antibody cocktail (20 mg/kg of c2H2 plus 20 mg/kg of c3C1) at 4 h.p.i., or 20 mg/kg of MAb c2H2, or the chimeric antibody cocktail at 24 h.p.i. For both experiments, all mice were euthanized at 3 d.p.i. and dissected to collect the lungs for viral RNA determination and histopathological examination.

Viral RNA was extracted from ground lung tissue with Trizol reagent (Invitrogen) and reverse transcribed using cDNA Synthesis Kit (Tiangen, China) according to the manufacturer's instructions. Real-time qPCR was performed using

SuperReal PreMix Plus SYBR Green kit (Tiangen) and the SARS-CoV-2 N gene-specific primers, as described above.

Mouse lungs were fixed in 4% paraformaldehyde solution. Tissue paraffin sections (2–4 μm in thickness) were stained with H&E. The slices were observed with Olympus microscope.

**SARS-CoV-2 S/Fab complex formation.** Purified 2H2/3C1 IgG was dialyzed against PBS (HyClone, USA), and then incubated with papain (300:1 W/W) in the presence of 20 mM L-cysteine and 1 mM EDTA for 3 h at 37 °C. The reaction was quenched by adding 20 mM iodoacetamide. Fab was purified by performing ion-exchange chromatography using a HiTrap DEAE FF column (GE Healthcare).

Purified SARS-CoV-2 S was incubated in a 1:4 molar ratio with 2H2 or 3C1 Fab on ice for 2 h before being subjected to purification by size-exclusion chromatography, using a Superose 6 increase 10/300 GL column (GE Healthcare) in 20 mM Tris-HCl pH 7.5, 200 mM NaCl, and 4% glycerol. The complex peak fractions were concentrated and assessed by SDS–PAGE and negative-stain electron microscopy (NS-EM).

**Negative-stain sample preparation, data collection, and initial model building.** For the NS sample, a volume of 5 μL of the S-2H2 sample was placed on a glow-discharged copper grid for 30 s. Excess sample on the grid was blotted off using filter paper, and a volume of 5 μL of 0.75% UF (Sigma-Aldrich) was added to wash the grid. After blotting, another volume of 5 μL of 0.75% UF was placed on the grid again for one minute to stain.

The S-2H2 sample was imaged on a Tecnai G2 Spirit 120 kV transmission electron microscope (Thermo Fisher Scientific) using an Eagle camera at a nominal magnification of 67,000× (yielding a pixel size of 1.74 Å). A total of 9884 particles were picked using EMAN2 (ref. [49]). All particles were extracted and subjected to reference-free 2D classification in Relion 3.1 (ref. [50]). Then good classes including 9305 particles were used to generate an initial model in Relion 3.1. For S-3C1 complex, the same procedure was adopted to generate an initial model from 94,606 cleaned up particles.

**Cryo-EM sample preparation and data collection for the S-2H2 and S-3C1 complexes.** An aliquot (~2.2 μL) of the S-2H2 sample was applied on a glow-discharged holey carbon grid (R1.2/1.3, 200 mesh; Quantifoil) or a graphene oxide-lacey carbon grid (300 mesh, EMR company). The grid was blotted with Vitrobot Mark IV (Thermo Fisher Scientific) and then plunged into liquid ethane cooled by liquid nitrogen. To handle the preferred orientation problem, for sample frozen using holey carbon grid, 0.05% octyl β-D-glucopyranoside (Sigma) or 0.1% polylysine (Polysciences) was added into the sample or applied on grid before freezing, respectively. The above-mentioned procedure was also followed to prepare the cryo-EM grids for the S-3C1 complex.

Movies for the cryo-EM samples were collected on a Titan Krios electron microscope (Thermo Fisher Scientific) operated at an accelerating voltage of 300 kV with a nominal magnification of 22,500× (Supplementary Table 1). The movies were recorded on a K2 Summit direct electron detector (Gatan) operated in the super-resolution mode (yielding a pixel size of 1.02 Å after two times binning) in an automatic manner using SerialEM[51]. Each frame was exposed for 0.15 s at the dose rate of 8 e$^-$/Å$^2$·s and the total accumulation time was 6.45 s, leading to a total accumulated dose of 49.6 e$^-$/Å$^2$ on the specimen (Supplementary Table 1).

**Cryo-EM 3D reconstruction.** For both datasets, the motion correction of each image stack was performed using the embedded module of Motioncor2 (ref. [52]) in Relion 3.1 (ref. [50]) and CTFFIND4 was used to determine CTF parameters before further data processing[53]. For the S-2H2 dataset, unless otherwise described, the data processing was performed in Relion 3.1 (ref. [50]). After automatic particle picking, manual selection, and multiple rounds of reference-free 2D classification, cleaned up particles remained for further reconstruction with the NS-EM map as initial model (Supplementary Fig. 9). After multiple rounds of 2D and 3D classifications, we obtained a S-2H2-F3a map from 37,641 particles and a S-2H2-F2 map from 17,819 particles. After CTF refinement and Bayesian polishing, the S-2H2-F3a and the S-2H2-F2 maps were refined to 3.8 and 4.3 Å resolution, respectively. The overall resolution was determined based on the gold-standard criterion using an FSC of 0.143. Moreover, the 3D classification also yielded a S-2H2-F1 map from 6382 particles and a S-2H2-F3b map from 8083 particles, which were further refined by homogeneous refinement and nonuniform refinement in cryoSPARC[54], respectively.

For the S-3C1 dataset, unless otherwise described, the data processing was mainly performed in cryoSPARC[54]. A total of 1,091,604 particles were picked from original micrographs, and all the particles were refined and re-centered against the NS-EM map as initial model in Relion 3.1 (Supplementary Fig. 11). We then loaded these particles into cryoSPARC for subsequent processing. After 2D classification, we obtained a dataset of 416,693 cleaned up particles. After heterogenous refinement, we obtained four classes, among which classes 1, 2, and 4 were further cleaned up by 2D classification and nonuniform refined to 3.0, 6.3, and 4.3 Å resolution corresponding to S-close, S-open, and S-3C1-F3b state, respectively. As for class 3, we performed multiple rounds of heterogenous refinement/nonuniform refinement, and eventually obtained another three distinct states, namely, S-3C1-F3a, S-3C1-F1, and S-3C1-F2

at 5.9, 7.5, and 5.6 Å resolution, respectively (Supplementary Fig. 11). The overall resolution was determined based on the gold-standard criterion using an FSC of 0.143.

**Pseudo atomic model building**. The homology model of 2H2 or 3C1 Fab was built through SWISS-MODEL webserver[55]. For the resolved series of S-2H2 and S-3C1 structures, we used the available open state model of SARS-CoV-2 S trimer (PDB: 6VYB)[12] as initial model, with the NTD domain being replaced by the counterpart in our recently better resolved SARS-CoV-2 S-closed structure using coot[14]. Then the models of individual subunits were fitted into the density map as rigid body using UCSF Chimera, and subsequently combined as a complete model[56]. For S-close and S-open states, the models were built based on the SARS-CoV-2 S-close and S-open structures from our recent study[14]. Subsequently, each of the models was refined against corresponding cryo-EM density map using Rosetta[57] then Phenix[58]. The final pseudo atomic models were validated using Phenix.molprobity command in Phenix.

UCSF Chimera and ChimeraX were used for map segmentation and figure generation[56,59].

**Statistical analysis**. All statistical analyses were performed using GraphPad Prism version 5.

**Reporting summary**. Further information on experimental design and research design is available in the Nature Research Reporting Summary linked to this article.

## Data availability

The authors declare that the data supporting the findings are available from the corresponding authors upon reasonable request. Cryo-EM maps determined in the SARS-CoV-2 S-2H2 dataset have been deposited at the Electron Microscopy Data Bank with accession codes EMD-30703, EMD-30704, EMD-30702, and EMD-30705, and associated atomic models have been deposited in the Protein Data Bank with accession codes 7DK5, 7DK6, 7DK4, and 7DK7 for S-2H2-F1, S-2H2-F2, S-2H2-F3a, and S-2H2-F3b, respectively. Cryo-EM maps determined in the SARS-CoV-2 S-3C1 dataset have been deposited at the Electron Microscopy Data Bank with accession code of EMD-30654, EMD-30651, EMD-30649, EMD-30642, EMD-30641, and EMD-30635, and related models have been deposited in the Protein Data Bank under accession code of 7DDN, 7DDD, 7DD8, 7DD2, 7DCX, and 7DCC for S-open, S-closed, S-3C1-F1, S-3C1-F2, S-3C1-F3a, and S-3C1-F3b, respectively. The sequences of 3C1-VH, 3C1-VL, 2H2-VH, and 2H2-VL have been deposited in GenBank with the accession codes MW271801, MW271802, MW271803, and MW271804, respectively. Source data are provided with this paper.

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

## Acknowledgements

We thank Dr. Xiaozhen Liang for providing K562 cell lines, Dr. Guangxun Meng for THP-1 cell line, Drs. Gary Wong and Jiaming Lan for codon-optimized SARS-CoV-2 RBD gene, and the staffs of the NCPSS Electron Microscopy facility, Database and Computing facility, and Protein Expression and Purification facility for instrument support and technical assistance. Z.H. was supported by grants from the Chinese Academy of Sciences (XDB29040300), from the Ministry of Science and Technology of China (2020YFC0845900), and from the Shanghai Municipal Science and Technology Major Project (20431900402). Y.C. was supported by grants from the CAS Pilot Strategic Science and Technology Projects B (XDB37040103), the National Basic Research Program of China (2017YFA0503503), the NSFC (31670754 and 31872714), the CAS Major Science and Technology Infrastructure Open Research Projects, the Program of Shanghai Academic Research Leader (20XD1404200), and the CAS-Shanghai Science Research Center (CAS-SSRC-YH-2015-01, and DSS-WXJZ-2018-0002). C.Z. was supported by the Youth Innovation Promotion Association of the Chinese Academy of Sciences (CAS). The BSL-3 lab of Fudan University was supported by Shanghai Science and Technology Committee and Project of Novel Coronavirus Research from Fudan University.

## Author contributions

Z.H., Y.C., Y.H.X., and Q.D. designed the study; C.Z., Y.F.W., Y.F.Z., C.X.L., C.J.G., S.Q. X., Y.L.W., Y.Z., Y.X.W., W.Y.H., X.Y.H., Y.Y., X.Y.Z., T.F.W., C.X., Q.H., S.T.W., Q.Y.Z., W.H.Q., J.K.Z., L.L.K., F.F.W., and H.K.W. performed experiments; all authors analyzed data; C.Z., Y.F.W., C.X.L., Y.C., and Z.H. created figures; Z.H., Y.C., C.Z., Y.F.W., and C.X.L. wrote the manuscript with help from all authors; and D.Q., D.L., and H.T. advised the researcher and reviewed the article.

## Competing interests

Z.H., C.Z., Y.L.W., and S.Q.X. are listed as inventors on pending patent applications for MAbs 2H2 and 3C1. The other authors declare that they have no competing interests.
