## [Peer Review File · Nature Communications]

REVIEWER COMMENTS

Reviewer #1 (Remarks to the Author):

Development and structural basis of a two-MAb cocktail for treating SARS-CoV-2 infections by Zhang et al.

This is a very detailed and important work to discover potential antibodies that can be used as therapeutics treatment of SARS-CoV2. The authors made mouse hybridomas, screened for their binding to SARS-CoV2 RBD, ability to compete binding with ACE2 receptor, their neutralization capability towards pseudovirus and live virus. They also did competitive binding with between these antibodies to each other in BLI and showed that the shortlisted antibodies can be classified into two groups – group 1 (3C1) and the group 2 (2H2, 2G3, 3A2, 8D3) and they have done a rough epitope mapping with different constructs containing different parts of the RBD substituted with that of SARS-CoV and showed that while 3C1 likely bind to the RBD core, the other antibodies to a peptide T470 to T478. To make an antibody cocktail, they further shortlisted 3C1 and 2H2 because 2H2 is highly neutralizing and 3C1 likely binds to a non-overlapping epitope. They then humanized these two antibodies and test the ability to neutralize in vitro and in vivo (mice with ACE2 receptor) for neutralizing activity. They further characterized structurally how these two antibodies individually bind to the S proteins by cryoEM and have observed various different structural states of binding. The work is important and maybe useful in producing prophylactic and therapeutic treatments. Overall the manuscript is well written and the experiments are well conducted.

Major comments

(1) 2H2 is a highly neutralizing antibody, while 3C1 is weakly neutralizing or non-neutralizing (Neut50 of 3.1ug/ml to live virus). It is not clear whether the addition of 3C1 to 2H2 in a cocktail improved anything, as 2H2 by itself is already highly neutralizing. The in vivo mice experiments (figure 3b) show the cocktail of c2H2/c3C1 in the delayed treatment (24hpi) reduces the viral load in the infected mice but there wasn't any c2H2 control to compare with.

(2) The authors mentioned that one possible important advantage of the c2H2/c3C1 cocktail is the virus is unlikely to undergo mutation to escape from these two antibodies at the same time. This is

a very important point to justify for the use of cocktail and so the authors should try to passage the virus in this cocktail for several passages to determine if escape mutants can be obtained.

(3) Structurally, individual Fab fragment of these antibodies complexed with S protein is done very well. This reviewer is wondering why the authors did not mix these two Fabs together to S protein to do another reconstruction.

(4) The best resolutions of these structures are 3.8 to 4Å resolution. The side-chain densities of these maps are probably poor and hence, side chains cannot be placed with high confidence. A distance of 4Å between side chains is therefore not appropriate for identification of interactions. The authors should analyze the structure by using <8Å distance cut off between c-alpha chains. The interactions between residues should be toned down to “likely” or “possible” interactions throughout the manuscript and in supplementary table 2.

Minor comments

(1) Page 6, line 166 delete “respectively”.

(2) Page 7, line 199 delete “respectively”.

(3) Figure 4h, maybe authors should compare the epitope bound by 2H2 and ACE2 by circling their borders onto the top view of RBM.

(4) Page 18, line 499 change “expose” to “exposure”.

(5) Make sure all “in vitro” and “in vivo” are all in italics.

Reviewer #2 (Remarks to the Author):

This is an impressive body of work in the breath and number of experiments used to identify, characterize and choose monoclonal antibodies against SARS-CoV-2, evaluate their efficacy in an animal model and then solve cryoEM structures of them. I particularly appreciated the multiple conformational states visible in the structures and the reasonable models of stepwise binding and resulting conformational adjustment to the RBDS. This is not the first work on this subject and since these are murine in origin, they are unlikely to become human therapeutics compared to others. Nonetheless, I was impressed by the depth and breadth of this study and the clarity with which the results were presented.

No specific edits.

Reviewer #3 (Remarks to the Author):

The authors provide an account of the discovery of two neutralizing mouse monoclonal antibodies, subsequently humanized, to non-competing epitopes on the receptor binding domain (RBD) of the S protein of SARS-CoV-2. They show the antibodies are potently neutralizing and protective in a mouse model of COVID. They further show the antibodies are effective in containing virus when given 24h after infection in the model. They did not observe ADE in the model. Finally, they carry out cryoEM studies on complexes of the Abs with S that lead them to propose stepwise allosteric changes in the S protein as the antibodies bind.

The paper is well written, thorough and solid. The novelty of the paper is somewhat restricted given the large number of publications on neutralizing antibodies to SARS-CoV-2 that have appeared in recent months. Nevertheless, the animal model data and cryoEM data are interesting and do add to understanding in the field.

I have only two significant comments. First, I do not think the claims with regard to the utility of the humanized antibodies are likely to be met given the available human antibodies from a number of sources. I believe human will be preferred over humanized where concerns about ADA will remain. Second, I find the cryoEM data very interesting but still I think the authors should be cautious in relating this to neutralization mechanism. Their studies are carried out on recombinant S protein and matters may be different on the virion surface. Given that both antibodies bind either to the ACE2 site or close to it, it is quite possible that the antibodies simply interfere with receptor binding and fusion as a neutralization mechanism.

Response to the reviewers' comments

Reviewer #1 (Remarks to the Author):

Development and structural basis of a two-MAb cocktail for treating SARS-CoV-2 infections by Zhang et al.

This is a very detailed and important work to discover potential antibodies that can be used as therapeutics treatment of SARS-CoV2. The authors made mouse hybridomas, screened for their binding to SARS-CoV2 RBD, ability to compete binding with ACE2 receptor, their neutralization capability towards pseudovirus and live virus. They also did competitive binding with between these antibodies to each other in BLI and showed that the shortlisted antibodies can be classified into two groups – group 1 (3C1) and the group 2 (2H2, 2G3, 3A2, 8D3) and they have done a rough epitope mapping with different constructs containing different parts of the RBD substituted with that of SARS-CoV and showed that while 3C1 likely bind to the RBD core, the other antibodies to a peptide T470 to T478. To make an antibody cocktail, they further shortlisted 3C1 and 2H2 because 2H2 is highly neutralizing and 3C1 likely binds to a non-overlapping epitope. They then humanized these two antibodies and test the ability to neutralize in vitro and in vivo (mice with ACE2 receptor) for neutralizing activity. They further characterized structurally how these two antibodies individually bind to the S proteins by cryoEM and have observed various different structural states of binding. The work is important and maybe useful in producing prophylactic and therapeutic treatments. Overall the manuscript is well written and the experiments are well conducted.

Response: Thanks for the positive overall evaluation.

Major comments

(1) 2H2 is a highly neutralizing antibody, while 3C1 is weakly neutralizing or non-neutralizing (Neut50 of 3.1ug/ml to live virus). It is not clear whether the addition of 3C1 to 2H2 in a cocktail improved anything, as 2H2 by itself is already highly neutralizing. The in vivo mice experiments (figure 3b) show the cocktail of c2H2/c3C1 in the delayed treatment (24hpi) reduces the viral load in the infected mice but there wasn't any c2H2 control to compare with.

Response: Indeed, 3C1 is a weak neutralizer. However, we did find that the addition of c3C1 to c2H2 in a cocktail resulted in slightly better neutralization in vitro than c2H2 alone (Fig. 2g and 2h, and Supplementary Fig. S7e). After receiving the review comments, we have performed another delayed treatment (24 hpi) experiment to compare the therapeutic efficacies of c2H2 alone and the c2H2/c3C1 cocktail, and

found that both c2H2 and cocktail treatments could significantly reduce viral loads in the infected mice as compared to the control (PBS) treatment. We should mention that the difference in viral load between the c2H2 and the cocktail groups was not statistically significant. This is not surprising, as the cocktail exhibited only slightly enhanced in vitro neutralization potency which may not readily transform into significantly improved protection in the mouse model used in the present study. The new data are presented in Figure 3b in the revised manuscript (also shown below for the convenience of the editor and reviewer).

Fig. 3b Therapeutic efficacy of MAb 2H2, c2H2 and/or the c2H2/c3C1 cocktail against SARS-CoV-2 infection. Upper left panel: study outline. Upper right panel: qRT-PCR analysis of viral RNA copies present in lung tissues after 3 days of infection. Lower panel: H&E staining of lung tissue sections at 3 dpi. qPCR results are expressed as viral RNA levels in different antibody treatment groups relative to that in the PBS control group. Statistical significance was indicated as follows: *, $P < 0.05$; ns, not significant.

(2) The authors mentioned that one possible important advantage of the c2H2/c3C1 cocktail is the virus is unlikely to undergo mutation to escape from these two antibodies at the same time. This is a very important point to justify for the use of cocktail and so the authors should try to passage the virus in this cocktail for several passages to determine if escape mutants can be obtained.

[redacted]

(3) Structurally, individual Fab fragment of these antibodies complexed with S protein is done very well. This reviewer is wondering why the authors did not mix these two Fabs together to S protein to do another reconstruction.

Response: Our cryo-EM study showed that, besides the ligand-free S proteins, there are four conformations for each of the S-2H2 and S-3C1 complexes, which are already quite complex to analyze. Moreover, we have mentioned that the RBDs of S trimer are very dynamic to coordinate the binding of 2H2/3C1 Fabs (Fig. S12h). If we mixed these two Fabs together with S protein to do reconstruction, the system could be much more conformationally and compositionally heterogeneous and would require humongous amount of cryo-EM data. However, due to the extremely tight Titan Krios machine time at the National Cryo-EM Facility (Shanghai) where we collected our data, it is unmanageable to collect enough dataset and to process such complex system within limited time. We highly appreciate the suggestion from the reviewer and will carry out such study in the near future; still, we feel this is beyond the scope of the current study.

(4) The best resolutions of these structures are 3.8 to 4Å resolution. The side-chain densities of these maps are probably poor and hence, side chains cannot be placed with high confidence. A distance of 4Å between side chains is therefore not appropriate for identification of interactions. The authors should analyze the structure by using <8Å distance cut off between c-alpha chains. The interactions between residues should be toned down to “likely” or “possible” interactions throughout the manuscript and in supplementary table 2.

Response: The point is well taken. We are aware that the local resolution in the RBD-Fab portion in our maps is not very high due to the intrinsic dynamic nature in these regions. We therefore followed the suggestion from the reviewer to tune down the tone and use “likely” or “possible” when describing the specific interactions in the revised manuscript. We also analyzed the structure by using <8 Å distance cutoff between main chains, and it appears that most of the interactions analyzed by <8 Å main chain distance cutoff or <4 Å side chain distance cutoff remain the same. In revised Table S2 and S3, we highlighted the interactions that fulfill both criteria, to indicate those interactions would have more confidence. Accordingly, we have now updated Fig. 4I-J and also modified the related text in the revised manuscript (lines 335-345 in P. 12). For the convenience of the reviewer and editor, the revised Table S2 and S3, and Fig. 4I-J were also shown below.

Table S2 Contacting residues (a sidechain distance cutoff of 4 Å) at the SARS-CoV-2 RBD/2H2 interfaces

SARS-CoV-2 S RBD	2H2
R403	L58, E59, S60
D405	L58

K417*	N57
V445	Q1
G446*	Q1
Y449*	M106
Y453*	Y53
L455*	Y53, L54
F456*	Y32
A475*	D30, S31
V483	R53, G54, G55
E484	W52, R53, N98, H102
G485	W52, D58
F486*	D98, D58
N487*	N95, N96, H102
C488	H102
Y489*	S31, Y32, N95, G100, A101, H102
F490	G100
Q493*	G99, G100, D105
G496*	S60
Q498*	S60, G61
N501*	S60
Y505*	L58, E59, S60, V62, P63, A64

Heavy chain

Light chain

* ACE2 binding sites

Residues in coral indicate interactions also fulfill the criterion of < 8 Å main chain distance cutoff

Table S3 Contacting residues (a sidechain distance cutoff of 4 Å) at the SARS-CoV-2 RBD/3C1 interfaces

SARS-CoV-2 S RBD	3C1
N501*	T30
G502*	Y53
V503	N31, G32, Y33, Y53, Y99
G504	Y33, S52, S54
Y505*	S54
Q506	N31, Y99
Y508	Y33, Y99
V433	R93
N437	Y99
A411	R93
Q414	R93
R403	S54, S56

D405	Y50, S52, S54, S56, Y58
R408	Y50, Y58, R93, Y94
Q409	Y58, R93
I410	R93
A372	N31, W50
F374	D32
S375	N92, R93
T376	N92, R93
F377	V29, N92
K378	I2, V29, Q90, N92, R93
C379	Q27
Y380	R93
S383	D28
P384	D28
T385	D28
Y369	D28, G30, G68
N370	N31

Heavy chain

Light chain

* ACE2 binding sites

Residues in coral indicate interactions also fulfill the criterion of $< 8 \text{ \AA}$ main chain distance cutoff

Fig. 4. Cryo-EM structures of the SARS-CoV-2 S trimer in complex with 2H2 Fab. (h) 2H2 Fab (left) and ACE2 (right, gold, PDB: 6M0J) share overlapping epitopes on RBM (second row) and would clash upon binding to the S trimer. (i and j) The involved regions/residues forming potential contacts between the light chain (in violent red, i) or heavy chain (in royal blue, j) of 2H2 and the RBD-1 of S-2H2-F3a. Asterisks

highlight residues also involved in the interactions with ACE2. Note that considering the local resolution limitation in the RBD-2H2 portion of the map due to intrinsic dynamic nature in these regions, we analyzed the potential interactions fulfill criteria of both $< 4 \text{ \AA}$ side chain distance cutoff and $< 8 \text{ \AA}$ main chain distance cutoff, which criteria were followed throughout.

Minor comments

(1) Page 6, line 166 delete “respectively”.

Response: Thanks for pointing it out. We have now deleted “respectively” as suggested.

(2) Page 7, line 199 delete “respectively”.

Response: We have now deleted “respectively”.

(3) Figure 4h, maybe authors should compare the epitope bound by 2H2 and ACE2 by circling their borders onto the top view of RBM.

Response: We have followed the suggestion from the reviewer to compare the binding epitopes of 2H2 and ACE2 in the top view of RBM and modified Fig. 4h accordingly (also shown above for the convenience of the editor and reviewers).

(4) Page 18, line 499 change “expose” to “exposure”.

Response: As suggested, “expose” has been changed to “exposure”.

(5) Make sure all “in vitro” and “in vivo” are all in italics.

Response: The suggested changes have been made throughout the manuscript.

Reviewer #2 (Remarks to the Author):

This is an impressive body of work in the breath and number of experiments used to identify, characterize and choose monoclonal antibodies against SARS-CoV-2, evaluate their efficacy in an animal model and then solve cryoEM structures of them. I particularly appreciated the multiple conformational states visible in the structures and the reasonable models of stepwise binding and resulting conformational adjustment to

the RBDS. This is not the first work on this subject and since these are murine in origin, they are unlikely to become human therapeutics compared to others. Nonetheless, I was impressed by the depth and breadth of this study and the clarity with which the results were presented.

Response: Thanks for the positive overall evaluation.

No specific edits.

Reviewer #3 (Remarks to the Author):

The authors provide an account of the discovery of two neutralizing mouse monoclonal antibodies, subsequently humanized, to non-competing epitopes on the receptor binding domain (RBD) of the S protein of SARS-CoV-2. They show the antibodies are potently neutralizing and protective in a mouse model of COVID. They further show the antibodies are effective in containing virus when given 24h after infection in the model. They did not observe ADE in the model. Finally, they carry out cryoEM studies on complexes of the Abs with S that lead them to propose stepwise allosteric changes in the S protein as the antibodies bind.

The paper is well written, thorough and solid. The novelty of the paper is somewhat restricted given the large number of publications on neutralizing antibodies to SARS-CoV-2 that have appeared in recent months. Nevertheless, the animal model data and cryoEM data are interesting and do add to understanding in the field.

Response: Thanks for the positive overall evaluation.

I have only two significant comments. First, I do not think the claims with regard to the utility of the humanized antibodies are likely to be met given the available human antibodies from a number of sources. I believe human will be preferred over humanized where concerns about ADA will remain.

Response: Thanks for the insightful comments. We agree that in general fully human antibodies will be preferred over humanized antibodies, however, many humanized antibodies have been commercialized, such as Palivizumab (a humanized MAb for treating RSV infection) and many anti-cancer MAbs, indicating such an approach is viable. We do understand that c2H2 and c3C1 described in this manuscript are mouse-human chimeras (containing the entire variable regions of murine origin) and for future human use the antibodies need to be further humanized by CDR grafting into human germlines to reach more than 90% sequences as human origin. To reflect

this point, we have added a statement (Page 17, lines 472-474), to read “We should point out that c2H2 and c3C1 are mouse-human chimeras and therefore for future human use the antibodies will need to be further humanized by grafting their CDRs into a suitable human MAb backbone. “.

Second, I find the cryoEM data very interesting but still I think the authors should be cautious in relating this to neutralization mechanism. Their studies are carried out on recombinant S protein and matters may be different on the virion surface. Given that both antibodies bind either to the ACE2 site or close to it, it is quite possible that the antibodies simply interfere with receptor binding and fusion as a neutralization mechanism.

Response: Thanks. The reviewer’s point about being cautious in relating our cryoEM data to neutralization mechanism is well taken. We have now modified the related statements in the Discussion. For example, “we propose a stepwise binding and neutralizing mechanism of 2H2/3C1 Fabs targeting the RBD domain of the SARS-CoV-2 S trimer “has been changed to “we propose a model of stepwise binding of 2H2/3C1 Fabs to the RBD domain of the SARS-CoV-2 S trimer” (please see page 18, line 513-514); and Figure 6 title “Fig. 6. A proposed stepwise binding and neutralizing mechanism of 2H2/3C1 Fabs targeting the RBD of SARS-CoV-2 S trimer.” has been changed to “A proposed model of stepwise binding of 2H2/3C1 Fabs to the RBD of SARS-CoV-2 S trimer.”

Reference

1. Baum A, *et al.* Antibody cocktail to SARS-CoV-2 spike protein prevents rapid mutational escape seen with individual antibodies. *Science*, (2020).

REVIEWERS' COMMENTS

Reviewer #1 (Remarks to the Author):

This reviewer is happy with the changes and there's no further comments.